



# Contribution of the 2DVD to the investigation of cloud microphysics during the MOSAiC and Cloudlab/PolarCAP campaigns

Tom Gaudek[1], Cristofer Jimenez[1], Kevin Ohneiser[1], Christopher Fuchs[2], Jan Henneberger[2], Johannes Bühl[1,3], Andi Klamt[1], Albert Ansmann[1], Ronny Engelmann[1], and Patric Seifert[1]

[1]Leibniz Institute for Tropospheric Research, Leipzig, Germany
[2]ETH Zürich, Zürich, Switzerland
[3]Harz University of Applied Sciences, Wernigerode, Germany

**Correspondence:** Patric Seifert (seifert@tropos.de)

**Abstract.**

In this study, the particle maximum diameter is introduced and evaluated as a new variable of the two-dimensional video disdrometer (2DVD). Vertically resolved remote-sensing measurements meanwhile allow to retrieve the microphysical properties of precipitation. However, opportunities for a direct evaluation of those retrievals are still lacking. One possible approach
is the ground-based observation of precipitation particles with in-situ sensors such as the 2DVD. In this context, the suitability of the 2DVD for contributing to cloud microphysics studies is being assessed. First, the retrieval of the particle maximum diameter as a new parameter is described, followed by an explanation about the procedure of the determination of dominating particle shapes done in this study. The capabilities of the 2DVD are demonstrated by means of measurements performed in a pre-alpine region of Switzerland which show that the instrument could detect signatures from cloud seeding experiments.
Moreover, ice crystal number concentration and, for the first time, mean maximum diameter derived from the remote-sensing based LIRAS-ice retrieval are evaluated against ground-based in-situ measurements from the 2DVD. In the frame of a case study from the Multidisciplinary drifting Observatory for the Study of Arctic Climate (MOSAiC) expedition in 2019, LIRAS-ice and 2DVD data were found to agree well during suitable meteorological conditions that allow to relate surface observations to the higher-level remote sensing measurements. This study shows that the maximum diameter from 2DVD observations
enhances the instruments capability to contribute to precipitation-related research.

## 1 Introduction

Cloud-microphysical processes remain a present-day object of investigation in atmospheric research (IPCC, 2021). The formation, development, and dissipation of clouds is influenced by the presence of aerosol particles (aerosol-cloud-interaction, ACI; Lohmann and Feichter, 2005; Forster et al., 2021), especially at mid-latitudes (Kanitz et al., 2011; Radenz et al., 2021). In air
masses which are saturated with respect to liquid water, aerosol particles may act as cloud condensation nuclei inducing cloud droplet formation. For temperatures below 0°C, some types of aerosol particles may additionally act as ice-nucleating particles initiating heterogeneous ice formation if the air is saturated with respect to ice (Hoose and Möhler, 2012; Kanji et al., 2017). Below approximately -37°C, this process may be accompanied by homogeneous freezing which does not require solid-state



aerosol particles as nuclei (Kärcher, 2002). However, cloud formation does not only depend on aerosol particles but also on
dynamics and interactions between the water species themselves (Guichard and Couvreux, 2017). Particularly in mixed-phase-
clouds (MPC), a variety of different hydrometeor types may occur simultaneously and interact with each other (Morrison
et al., 2011). The Wegener-Bergeron-Findeisen process (Wegener, 1911; Bergeron, 1935; Findeisen, 1938) is one example for
such interactions and dominates cloud evolution in the mid-latitudes (Boucher et al., 2013). This process allows ice crystals to
grow at the expense of liquid water. Therefore, MPC are not in a thermodynamic equilibrium and to remain persistent, certain
conditions are required (Morrison et al., 2011). For the investigation of cloud processes, the shape of ice crystals may hold
important information about the processes involved in the microphysical evolution of the precipitation (Pasquier et al., 2023).
When pristine ice crystals are growing, the resulting shape and size depend on the temperature and the supersaturation of the
ambient air with respect to ice (Bailey and Hallett, 2009). Therefore, the shape and size of ice crystals are important quantities
for the investigation of cloud microphysical processes and thus need to be detected.

Clouds and ACI can be investigated in the field, for example, with aimed remote-sensing measurements. For this purpose,
a synergy of various instruments with different advantages is often necessary to investigate the current atmospheric conditions
and to detect atmospheric processes more accurately. On the one hand, cloud radars are suitable for observing precipitation and
clouds containing ice particles (Kollias et al., 2007). Aerosol particles and small liquid-water cloud droplets, on the other hand,
can be better observed with lidar (Jimenez et al., 2020). Remote sensors with polarisation capability can help to further identify
predominant particle types, because mainly the hydrometeor shape determines which fraction of the incoming polarised radi-
ation is depolarised when scattered back towards the sensor (Myagkov et al., 2016). If instruments are operated that are able
to measure Doppler velocities, vertical wind information may additionally help to understand the impact of dynamics on the
cloud system. By means of retrieval techniques, cloud properties such as the ice crystal number concentration (ICNC; e.g. Bühl
et al., 2019) or the dominating crystal shape (e.g. Myagkov et al., 2016) can be retrieved. To specifically investigate clouds
and ACI under certain environmental conditions, an excellent opportunity is offered by cloud seeding experiments such as the
ones conducted in the frame of the CLOUDLAB project led by ETH Zürich (Henneberger et al., 2023). By releasing aerosols
into clouds, the cloud properties (i.e. concentrations of cloud water and cloud ice) are changed and this transformation can be
observed with extensive remote-sensing and in-situ measurements.

In-situ measurements of ground precipitation can be very valuable for the investigation of cloud-microphysical processes
for different reasons. On the one hand, they may deliver missing puzzle pieces for closure studies. On the other hand, they
can be used to evaluate remote-sensing retrievals by comparing measured and retrieved or assumed quantities of precipitation
properties like particle size distribution (PSD) or dominating particle shape. For precipitation research, a number of in-situ
sensors which observe falling hydrometeors in at least two dimensions were used in recent studies. Examples include the Two-
Dimensional Video Disdrometer (2DVD; Ansmann et al., 2025), the Precipitation Imaging Package (PIP; Vogl et al., 2022),
the Multi-Angle Snowflake Camera (MASC; Leinonen et al., 2021), or the Video In Situ Snowfall Sensor (VISSS; Maahn
et al., 2024). The 2DVD proved advantageous for campaign deployment, for example, due to the easy instrument setup and
data analysis compared to other measurement systems. Although the device was originally designed to measure liquid-droplet
properties (Kruger and Krajewski, 2002), 2DVD research focused especially during the last two decades on solid or mixed-





phase precipitation. Studies were undertaken, for example, about more suitable matching algorithms (Hanesch, 1999; Huang
et al., 2010; Grazioli et al., 2014; Bernauer et al., 2015; Helms et al., 2022) or automatized particle classification (Grazioli et al.,
2014; Gavrilov et al., 2015; Lee et al., 2015; Bernauer et al., 2016). The determination of dominating precipitation particle type
and shape is in particular helpful for comparing and evaluating remote-sensing-based retrievals as well as model simulations
of cloud and precipitation parameters (Trömel et al., 2021). A comprehensive 2DVD-related literature review can be found
in a recent Master's thesis (Gaudek, 2024). One shortcoming of the 2DVD is that the manufacturers software only delivers
the volume-equivalent diameter and the volume itself as size variables. However, other in-situ sensors and remote-sensing
retrievals often use the longest particle extent instead. This problem will be tackled in this work by introducing the maximum
diameter $d_{\max}$ as a new variable that can be calculated from 2DVD measurement data. Moreover, the contribution of the 2DVD
to the investigation of cloud microphysical processes is evaluated in this work in different ways.

In this paper, data from two measurement campaigns are used. The campaigns and operated instruments are presented in
Sect. 2 with a special focus on the 2DVD. In Sect. 3, $d_{\max}$ as a new 2DVD variable is introduced, the 2DVD's ability to
distinguish different particle shapes is tested, and the remote-sensing retrieval LIRAS-ice (LIdar RAdar Synergy – retrieval of
ICE microphysical properties) is explained. In Sect. 4, results of the 2DVD calibration, outcomes of the distinction of different
particle shapes by the 2DVD, measurement results from cloud seeding experiments, and findings from the combined LIRAS-
ice retrieval and 2DVD measurements are presented. The results are discussed in Sect. 5. A summary and conclusions are given
in Sect. 6.

## 2 Measurement campaigns and operated instruments

### 2.1 Cloudlab and PolarCAP

One part of the data which are analysed in this work was collected during a measurement campaign at Rapier-Platz near Eriswil,
Switzerland (47.0705°N, 7.8729°E, 921 m above sea level) from December 2022 to March 2023. Rapier-Platz is located in the
pre-alpine Swiss middleland between the Jura and the Alps mountains. One typical meteorological scenario during wintertime
is characterised by below-zero temperatures and the presence of steady winds from north-east called Bise (Wanner and Furger,
1990), often accompanied by supercooled stratus clouds which are suitable as natural cloud laboratory. The observations were
conducted within the framework of the two collaborating research projects Cloudlab of ETH Zürich (Henneberger et al., 2023)
and PolarCAP (Polarimetric Radar Signatures of Ice Formation Pathways from Controlled Aerosol Perturbations; Ohneiser
et al. (2025)) of Leibniz Institute for Tropospheric Research (TROPOS). Cloudlab in general aims at a better understanding
of cloud physics, the investigation of stratus formation and dissipation, and the validation and improvement of the cloud
microphysics scheme of the icosahedric nonhydrostatic (ICON) weather forecast model (Zängl et al., 2014; Omanovic et al.,
2024, 2025). In order to achieve these goals, a variety of field measurements, laboratory experiments, and modelling studies
were conducted. More specifically, seeding experiments in mixed-phase stratus clouds by means of uncrewed aerial vehicles
(Miller et al., 2024a), accompanied by remote sensing and in-situ observations provided new insights in cloud microphysical
processes (Miller et al., 2024b; Ramelli et al., 2024; Fuchs et al., 2025). Those observations were complemented by extensive,




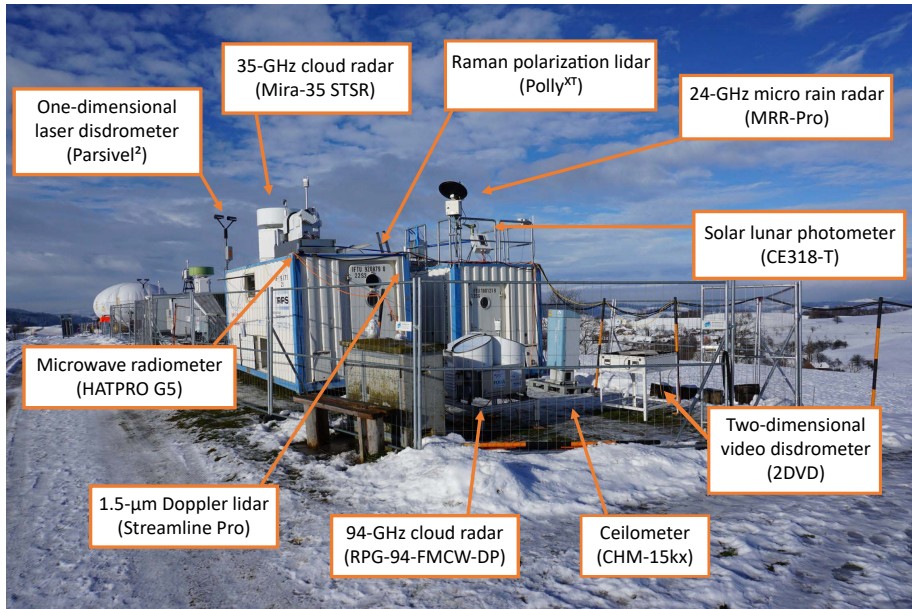

**Figure 1.** Setup of LACROS at Rapier-Platz, Switzerland, including instrument designations. Further instruments of ETH Zürich and METEK GmbH can be seen on the left in the background.

mainly remote-sensing-based measurements with the Leipzig Aerosol and Cloud Remote Observations System (LACROS; Bühl et al. (2013); Myagkov et al. (2016); Radenz et al. (2021); Teisseire et al. (2024); Ohneiser et al. (2025)) in the frame of PolarCAP. Figure 1 shows the setup of LACROS during the Cloudlab campaign. An overview of all LACROS instruments

which are relevant for this study is given in Table 1. Besides the support of Cloudlab with field observations and expertise, the PolarCAP aimed on the investigation of MPC development as well as the evaluation of MPC retrievals based on remote sensing. For the characterisation of MPCs, cloud-Doppler radar and (Doppler) lidar techniques were applied and Cloudlab in-situ measurements and seeding experiments were used. Moreover, simulations with numerical cloud-resolving models were conducted. Due to the variety of instruments, the measurement location and its duration, the presented field campaign offered

a unique opportunity to compare precipitation measurements of several remote sensing and in-situ instruments. In particular, the performance of the 2DVD under different conditions was evaluated in this work. Simultaneous measurements with the holographic in-situ sonde HOLIMO of ETH Zürich (Sect. 2.4) and remote-sensing instruments such as the 35 GHz and 94 GHz cloud Doppler radar contribute to the 2DVD studies.

## 2.2 MOSAiC

For the results presented in Sect. 4.4, 2DVD and remote sensing data from the Multidisciplinary drifting Observatory for the Study of Arctic Climate (MOSAiC) expedition (Shupe et al., 2020, 2022) were used. During the campaign, which took place from September 2019 until October 2020, the German research vessel *Polarstern* drifted, for the most time frozen in the sea





**Table 1.** Overview of the instrumentation which is relevant for this study. The instruments were either part of the mobile platform LACROS (1), OCEANET (2), or AMF-1 (3), except for the holographic imager HOLIMO (Ramelli et al., 2020) which was developed by ETH Zürich (see Sect. 2.4).

| Instrument | Type | Atmospheric parameters | Resolution |
|---|---|---|---|
| Polly[XT 1,2] | Multiwavelength Raman polarisation lidar, pointed 5° off-zenith | Particle backscatter and extinction coefficient; linear depolarisation ratio; water vapour mixing ratio | 7.5 m; 30 s |
| Mira-35[1] | 35 GHz scanning cloud radar | Vertical structure, boundaries and vertical-velocity dynamics of clouds and precipitation; contributes to cloud liquid and ice water profiles | 30 m; 3.5 s |
| RPG94_LACROS[1] | 94 GHz vertically pointing cloud radar | Vertical structure, boundaries and vertical-velocity dynamics of clouds and precipitation; profiles of cloud liquid water and ice water content | 15 – 30 m; 1 – 30 s |
| KAZR[3] | 35 GHz Doppler cloud radar | Radar reflectivity, mean Doppler velocity, Doppler spectrum width | 30 m; 2 s |
| Weather station[1] | Vaisala WXT536 | Atmospheric pressure, temperature, relative humidity, surface wind, rain rate | $\pm$ 3% at 10 m s$^{-1}$ $\pm$ 1 hPa |
| 2DVD[1,2] | Two-dimensional video disdrometer | Precipitation rate, size distribution, fall velocity, particle imaging | height < 0.17 mm*; width < 0.17 mm*; uncertainty of $\nu$ < 4%* * at $\nu$ < 10 m s$^{-1}$ |
| HOLIMO | Holographic imager | phase resolved size distribution, particle image, ice crystal habit | 6 µm – 2 mm |

ice, for the whole winter period in the high Arctic to enable the investigation of the central Arctic as a hotspot of global climate change. Onboard was the meteorological mobile research platform OCEANET (Griesche et al., 2020; Engelmann et al., 2021;
Ohneiser et al., 2021; Jimenez et al., 2025), which, as LACROS, housed a Raman lidar Polly[XT] as well as the 2DVD. Moreover, a 35 GHz Doppler cloud radar (Ka-band ARM Zenith Radar, KAZR) was deployed next to OCEANET on the Atmospheric Radiation Measurement (ARM) mobile facility 1 (AMF-1) of the US Department of Energy (ARM, 2025). Data of those instruments were used for the case study presented in Sect. 4.4. On 10 November 2019, for which the measurement data have been analysed, *Polarstern* was approximately located at 85.8°N, 116.1°E.

## 2.3 Two-dimensional video disdrometer (2DVD)


The 2DVD is a ground-based precipitation gauge which detects single precipitation particles within a certain measuring area. Originally designed to measure rain drop size distributions, the first instruments of this kind operated since 1996 (Schönhuber et al., 2007). The investigation of solid hydrometeors with such devices has been subject of research in recent years (e.g. Bernauer et al., 2015; Raupach et al., 2017; Zhang et al., 2021). The instrument which was operated during PolarCAP was





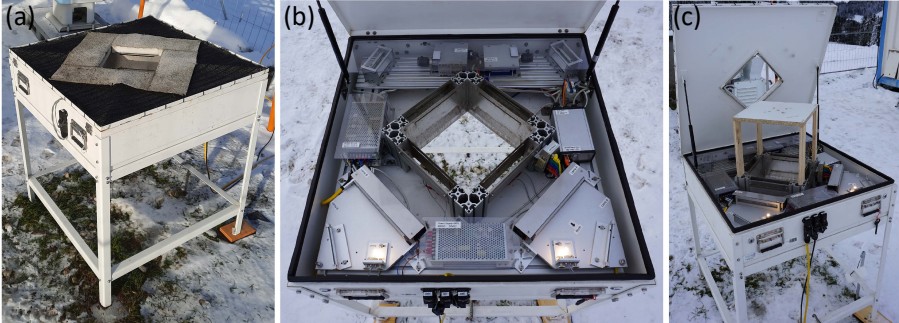

**Figure 2.** The 2DVD at the measurement site (a) in operating mode, (b) with open lid, and (c) during the calibration with calibration pattern.

developed by Joanneum Research, Graz, Austria. Table 1 shows the specifications of the 2DVD. It should be mentioned that the horizontal wind measured by the weather station mounted on the HATPRO G5 on the container roof during PolarCAP/-Cloudlab, and by the ultrasonic anemometer positioned next to the 2DVD during MOSAiC, did not exceed $10\,\mathrm{m\,s^{-1}}$ during the conduction of any 2DVD measurements presented in this work, unless otherwise stated.

### 2.3.1    Measurement principle and data processing

The schematic setup of the 2DVD is shown in Fig. 3. Two high-power lamps (halogen or LED) each continuously emit light which is reflected by a mirror onto a Fresnel lens. There, the light is diverged and travels through a slit plate so that an illuminated horizontal plane is created by each lamp. Those two planes have a height difference of around 6 mm. The area where the two sheets overlap is the so-called virtual measuring area and has a size of about 10 cm x 10 cm. After travelling through another slit plate, the light is then reflected by another mirror towards the respective line camera.

Regarding precipitation measurements, the instrument is designed advantageous for further reasons. The virtual measuring area is much smaller than the opening of the housing (25 cm x 25 cm) so that possible splashes of droplets hitting the housing would not be processed. Moreover, the lid is heated which avoids that snow accumulates on the instrument and gets blown into the virtual measuring area by wind gusts.

    If a hydrometeor falls through the virtual measuring area, its shape automatically gets reconstructed. In a first step, this is
done for the video signal of each camera separately. The photo detectors of each line camera generate an electronic signal at incident light. This signal is continuously compared against a threshold level. If the measured value is below this threshold, the corresponding pixel is recognised as shaded. By assembling the shaded pixels of consecutive time steps, a one-dimensional particle shape is finally reconstructed.

    In a second step, a matching algorithm is applied to merge the information for the same particle from both cameras. For this
purpose, several criteria that a pair of images needs to fulfil are considered. A time period is defined in which a particle needs to reach the lower light plane after it entered the upper light plane of the instrument. This time frame should be adapted to the relationship between particle size and fall velocity. In case of liquid droplets, this relation is well-known (e.g. Atlas et al., 1973).





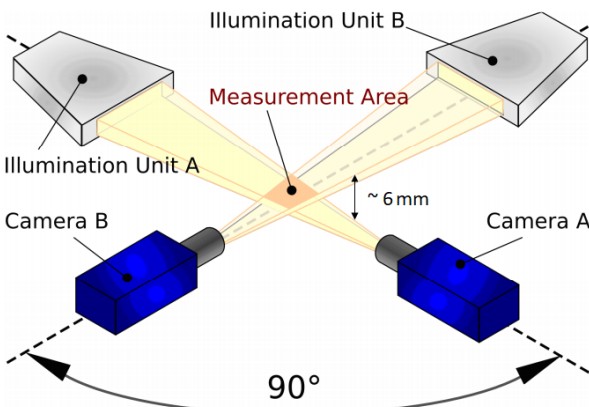

**Figure 3.** Measurement principle of the 2DVD. The illuminated planes are horizontally aligned with a vertical distance of around 6 mm while the respective camera perspectives differ by 90°. The measuring area is around 10 cm x 10 cm in size (adapted from Schönhuber et al., 2016).

For solid and mixed-phase hydrometeors, terminal velocities of particles of the same size can vary much more, depending on the particle type, shape, and, if applicable, the degree of riming (Locatelli and Hobbs, 1974; Barthazy and Schefold, 2006).

Thus, the time period to fall from the upper to the lower light plane should be set more broad in case of solid or mixed-phase precipitation. Moreover, droplets show regular symmetries which can be assumed by the matching algorithm. Solid particles, on the other hand, may appear with much more complex shapes. Matching algorithms for solid or mixed-phase precipitation have been continuously developed during the last years (Hanesch, 1999; Huang et al., 2010; Grazioli et al., 2014; Bernauer et al., 2015). For those reasons, the 2DVD's manufacturer provides two different matching algorithms for liquid and solid or

mixed-phase precipitation, measurements presented in this work have been processed with the algorithm for solid and mixed-phase precipitation. After the application of such an algorithm, single hydrometeors are identified and information about their properties are retrieved (see Sect. 2.3.3). Particles that are only seen in one camera or whose information of both cameras could not be merged are not further processed.

### 2.3.2 Calibration procedure

In order to calibrate the 2DVD, the following three main steps have to be performed. In a first step, the video levels of the live signals need to be adjusted so that the signal of each pixel is between two threshold values. The signal adjustment is done by aligning the optical elements of the 2DVD such as the lamps and mirrors. Once the signals are within the right intensity range, the distance of the two light planes needs to be measured precisely in a second step. For this purpose, steel spheres with a diameter of 10 mm are thrown through each hole of a calibration pattern (Fig. 2c). With this measurement as

an input, a programme then calculates the plane distance for every corner of the virtual measuring area. The resulting values need to be adjusted in a configuration file before further measurements are taken. Eventually, correction factors included in





the configuration file need to be evaluated. Therefore, a set of calibration steel spheres of 10 different diameters ranging from 0.5 mm to 10 mm are part of the 2DVD equipment. Ideally, more than 100 spheres of each diameter must be thrown through the virtual measuring area. If the measured diameters are in good agreement with the given diameters of the spheres, the calibration

is finished. If the agreement is poor over a series of repetitions of the calibration, the manufacturer of the 2DVD, Joanneum Research, can provide correction factors adapted to the taken measurements.

### 2.3.3 Standard 2DVD deliverables

By means of software which was delivered from the 2DVDs manufacturer, a set of parameters is determined for each single hydrometeor:

a. The equivalent spherical diameter $d_{\text{eq}}$ (mm) equals the diameter of a sphere with the same volume as the respective hydrometeor.

b. The vertical velocity $\nu$ (m s$^{-1}$) is determined by the time required by the hydrometeor to fall from the upper to the lower light sheet.

c. The particle volume $V$ (mm$^3$) is calculated by splitting the hydrometeor into horizontal slices whereas each slice is

assumed to have an area of an ellipse and the height of one line. $V$ is the sum of all slice volumes (Kruger and Krajewski, 2002).

d. The effective measuring area $A_{\text{eff}}$ (mm$^2$) is defined as the part of the virtual measuring area in which a hydrometeor with a certain size could have been detected. Between the detected particle and the borders of the virtual measuring area, there must be at least one non-shadowed pixel which reduces $A_{\text{eff}}$ the more the larger the particle is.

e. The oblateness $O$ (without unit) is defined as the ratio of particle height and particle width. The manufacturers software calculates $O$ from information of both cameras A and B as follows:

$$O = \sqrt{\left(\frac{height_{\text{A}}}{width_{\text{A}}}\right)\left(\frac{height_{\text{B}}}{width_{\text{B}}}\right)} \tag{1}$$

$width_{\text{A}}$ and $width_{\text{B}}$ represent the width of the widest scan line of the respective camera (Kruger and Krajewski, 2002). $O$ is an important shape parameter which can be helpful for the detection of different ice crystal shapes. However, for

solid particles, the applied method to calculate $O$ can be highly error-prone, especially at horizontal winds that tilt falling hydrometeors.

For the display of single particles, an own programme was written.



### 2.3.4 Precipitation properties

From the parameters of single particles, precipitation properties can be calculated. The following features were used for pre-
cipitation characterisation in this work:

a. Particle number concentration

The particle number concentration ($N$) describes the number of detected hydrometeors per unit of air volume:

$$N = \frac{1}{\Delta t} \sum_{j=1}^{M} \frac{1}{A_{\text{eff},j} \cdot 10^{-6} \cdot \nu_j} \qquad \left[\text{m}^{-3}\right] \tag{2}$$

with $M$ as number of particles during a time period $\Delta t$ (s) and $A_{\text{eff},j}$ (mm²) and $\nu_j$ (m s⁻¹) as effective measuring area
and vertical velocity, respectively, of particle $j$.

b. Precipitation rate $R$

The 2DVD is an advantageous instrument to determine precipitation rates under different weather conditions because
hydrometeors of various types can be well observed. $R$ of the 2DVD for liquid precipitation is calculated as follows:

$$R = \sum_{j(t_0)=1}^{j(t_0+\Delta t)} V_j \frac{3600}{\Delta t} \frac{1}{A_{\text{eff},j}} \qquad \left[\text{mm h}^{-1}\right] \tag{3}$$

with the integration time interval $\Delta t$ (s) and the effective measuring area $A_{\text{eff},j}$ (mm²) and volume $V_j$ (mm³) of each
particle $j$. In case of solid precipitation, the liquid equivalent volume $V_{\text{eq},j}$ of particles must be taken into account for
the calculation of $R$. For ice, a density $\rho_{\text{ice}}$ of 0.9 g cm⁻³ can be assumed which yields

$$V_{\text{eq},j} = V_j \frac{\rho_{\text{ice}}}{\rho_{\text{H}_2\text{O}}} \qquad \left[\text{mm}^3\right] \tag{4}$$

with $\rho_{\text{H}_2\text{O}} = 1 \, \text{g cm}^{-3}$ and eventually

$$R = \sum_{j(t_0)=1}^{j(t_0+\Delta t)} V_{\text{eq},j} \frac{3600}{\Delta t} \frac{1}{A_{\text{eff},j}} \qquad \left[\text{mm h}^{-1}\right] \tag{5}$$

However, especially for large snow particles, the exact determination of the particle volume is challenging because the
delicate structure of ice crystals can not be sufficiently resolved. For this purpose, studies about the snow bulk density in
dependence of particle size by means of 2DVD measurements were conducted (Brandes et al., 2007; Zhang et al., 2021)
and different density–size relations were determined. However, this topic requires further research and is not discussed
further in this work.



Precipitation can also be analysed by establishing time series of particle properties such as $d_{\mathrm{eq}}$, $V$, $O$, or $\nu$. For example, trends in precipitation evolution may become visible. Particularly with regard to the Cloudlab/PolarCAP campaign, cloud seeding signatures on the ground may be detected.

Moreover, the relation of variables mentioned in Sect. 2.3.3 to each other often indicates different particle types. Especially
investigating the relationship between the diameter and oblateness $O$ or the diameter and vertical velocity $\nu$ has proven to be particularly effective for precipitation analysis.

## 2.4 HOLIMO

The HOLographic Imager for Microscopic Objects (HOLIMO) is a holographic imager developed by ETH Zürich (Ramelli et al., 2021). With this instrument, cloud particles with diameters between 6 μm and 2 mm of a well defined sample volume of
12 cm³ can be imaged two-dimensionally. From the observations, the size spectrum and concentrations of cloud particles as well as the water content of clouds can be calculated. Moreover, for observed solid hydrometeors, different particle types such as large rimed particles, columnar crystals, or irregular particles can be distinguished from each other (Zhang et al., 2024). HOLIMO can either be operated on the ground or attached to slowly moving aircraft, such as helikites or ropeways. The instrument is compatible with the tethered balloon system HoloBalloon (Ramelli et al., 2020) which operated during the Polar-
CAP/Cloudlab campaign. This system allows the in-situ measurement of vertical profiles up to 1 km above the ground which enables detailed cloud characterisation. Furthermore, it may observe the changes of cloud particle properties during seeding experiments. Ground-based HOLIMO measurements next to the 2DVD enable the complementary detection of particles below the 2DVD's minimum resolution.

## 3 Methods

### 3.1 Maximum Feret diameter from 2DVD observations

Because the hydrometeors detected by the 2DVD are observed from only two perspectives, the derivation of the longest extent of a particle is challenging. The longest distance between two points of a geometrical two-dimensional shape is defined as Feret diameter. For each hydrometeor detected by the 2DVD, two images in the form of binary pixel information exist for the two camera perspectives. For each image, the Feret diameter is calculated and the larger of the two Feret diameters is defined
as $d_{\mathrm{max}}$ in this study. The so called minimum Feret diameter (or minimum Feret width), on the other hand, describes the minimum distance between two parallel tangents that can be drawn around the outline of each 2D image of the particle. The smaller of the two Feret diameters ($d_{\mathrm{max},2}$), as well as the minimum Feret diameters from each perspective ($d_{\mathrm{min}}$ and $d_{\mathrm{min},2}$), are additional variables retrieved by a newly developed algorithm, which is briefly described below. The introduced variables are illustrated on an exemplary dendrite in Fig. 4.

Due to varying positions in the measurement area and varying fall velocities, the pixel width and height can vary across hydrometeors but are constant within each binary image. The pixel width is usually by many times larger than the pixel height.





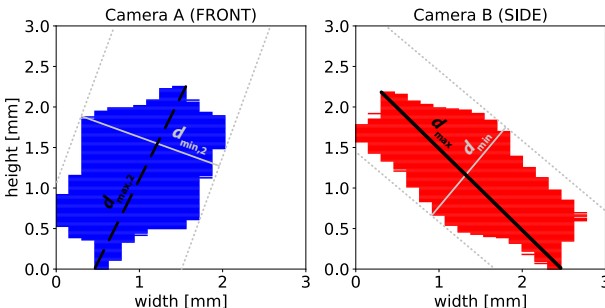

**Figure 4.** Visualisation of $d_{\max}$ (thick red line) for a dendrite-like particle. Thinner lines indicate further retrievable variables.

For the calculation of the Feret diameter for each camera perspective, an original two-dimensional binary pixel array is first obtained from an ASO (ASCII Standard Output) file generated by software of the 2DVD's manufacturer. From this input pixel array, a new binary array is created in which the pixel height remains unchanged, while the new pixel width equals the pixel height, resulting in square pixels. These square pixels allow an efficient calculation of the Feret diameter using the Python *feret* module (Feret, 2022).

The accuracy of $d_{\max}$ mainly depends on the particle orientation relative to the camera perspectives. The largest possible difference between $d_{\max}$ and the actual maximum extent of a particle occurs when a very thin and narrow particle such as a needle is aligned horizontally and in a way that the maximum diameter is oriented at 45° to both camera angles. Then the particle is seen from both cameras with a width $w$ and the true maximum particle extent $d$ equals $\sqrt{w^2 + w^2}$. Thus, the accuracy of $d_{\max}$ can be calculated by $\frac{w}{d} = \frac{w}{\sqrt{2w^2}} = \frac{w}{\sqrt{2}w} = 0.707$ which equals 70.7%. Since the 2DVD can fully capture the particle height and particle orientation and shape vary, the accuracy of $d_{\max}$ is typically higher. Another uncertainty source is the fact that camera pixels do not need to be fully blocked by the hydrometeor in order to be defined as 'shaded', only the voltage thresholds need to be met. This means that partially blocked pixels are often defined as 'shaded' but as the Feret diameter method accounts for the outermost pixel edge, this effect can lead to an overestimation of $d_{\max}$. However, the underestimation due to particle orientation relative to the camera perspectives and the overestimation due to partially covered pixels reduce the discrepancy of $d_{\max}$ from the true maximum extent. Due to the dependency on particle shape, orientation, and horizontal wind speed, the quantification of a potential mean accuracy of $d_{\max}$ is challenging.

$d_{\max}$ has some advantages over the previously used $d_{\mathrm{eq}}$. On the one hand, the maximum diameter of precipitating hydrometeors can be retrieved from other in-situ instruments such as the Video In-Situ Snowfall Sensor (VISSS, Maahn et al., 2024) which would allow a better instrument comparability. But also remote-sensing-based retrievals (e.g. Bühl et al., 2019) allow the determination of particle diameters, for which reason $d_{\max}$ is better suited when it comes to closure studies, for example. The combination of maximum and minimum Feret diameter would further allow the definition of a new aspect-ratio-like particle property which is not as sensitive to the canting angle as $O$ (see Section 2.3.3).



## 3.2 Dominating particle shapes in 2DVD data

In this work, dominating ice crystal shapes were identified by investigating relationships between $d_{max}$ and $\nu$ or $O$. Different particle shapes can typically be identified as separate clusters in the corresponding diagrams. Column-like particles have smaller $O$ and $\nu$ than plate-like particles, for example. The examination of recreated two-dimensional binary images of randomly selected particles from each cluster eventually reveals the respective shape. However, this method is limited to hydrometeors which appear as more than only five to six horizontal pixels in the 2DVD data. In case of very small particles which diameters below 1 mm, clusters might still be identifiable but the respective shape cannot be examined unequivocally. Former studies have shown, that automated particle typing from 2DVD data can work well, depending on the given particle classes (Grazioli et al., 2014; Gavrilov et al., 2015; Lee et al., 2015; Bernauer et al., 2016).

## 3.3 Ice crystal number concentration and size from remote sensing

Examining the ice crystal size and number concentration (ICNC) in clouds or for precipitating particles without in-situ measurements is challenging. However, Bühl et al. (2019) demonstrated that the ICNC can be derived by means of synergistic observations from lidar and cloud Doppler radar. The basic principle of this method can be explained as follows. For each height bin, the Doppler spectrum of radar reflectivity factor is determined from the cloud radar measurements while the optical extinction coefficient of the ice particles is derived from the lidar observations. In the next step, an algorithm retrieves the ice crystal size distribution which resembles best the observed radar Doppler spectrum of radar reflectivity factor and lidar optical extinction coefficient. Thus, an important constraint arises from the dominating crystal shape, which determines the mass- and area–size relationships required to compute the radar and lidar properties. This dominating crystal shape has to be presumed in advance. The mean crystal size is derived from the radar/lidar ratio and by scaling the size distribution to the particle extinction coefficient, the ICNC can eventually be computed. As the described method makes use of lidar-radar-synergy for deriving ice particle properties, it is further referred to as LIRAS-ice in this work.

For the successful application of LIRAS-ice, a number of meteorological and technical conditions must be met. First of all, both cloud radar and lidar measurements need to be available. The snow protection window of the lidar must not be significantly covered by snow as this would attenuate the laser beam and the backscattered radiation. Moreover, liquid droplets strongly attenuate the lidar backscatter signal and thus, the laser beam cannot sufficiently penetrate liquid or mixed-phase clouds. Therefore, no low clouds that contain liquid droplets should be present.

While it was already applied to long-term observations of mixed-phase and ice clouds (Jimenez et al., 2025), the LIRAS-ice approach has only partly been evaluated against in-situ observations yet (Ansmann et al., 2025). In this work, for the first time, retrieved mean maximum diameter and ICNC have both been compared with $d_{max}$ and $N$ from the 2DVD, respectively. For LIRAS-ice, the dominating shape observed by the 2DVD was used. Furthermore, the retrieval was executed for the same case with further assumed shapes for a more comprehensive evaluation. The search for time periods which are suited to conduct such case studies require further requirements besides those for a successful LIRAS-ice application. Only time periods were considered for which only one specific ice crystal shape could be identified by the 2DVD. Therefore, precipitation particles





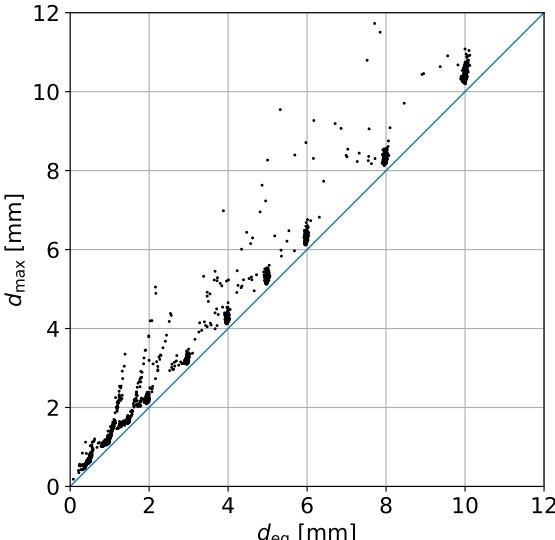

**Figure 5.** Comparison of $d_{eq}$ and $d_{max}$ of the set of steel spheres that were used for the 2DVD calibration on 12 December 2022.

were required to be larger than a minimum $d_{max}$ of approximately 1 mm, depending on the shape. For those reasons, the choice of suitable time periods was very much limited. One exemplary case study which met the requirements the most is presented in this work.

## 4 Results

### 4.1 Calibration of the 2DVD

At the beginning of the PolarCAP/Cloudlab campaign at Rapier-Platz, Switzerland, on 12 December 2022, the 2DVD was calibrated following the procedure described in Sect. 2.3.2. To evaluate $d_{max}$, the dataset of the measured calibration spheres was used and $d_{max}$ values were compared to the corresponding $d_{eq}$. The result of the calibration experiment is presented in Fig. 5. Whereas the 10 different size classes of the calibration spheres can well be identified, $d_{max}$ overestimates the true diameter by around 0% to 10% in case of diameters of 2 mm and larger. $d_{eq}$ agrees better than $d_{max}$ with the true diameter of the steel spheres. However, this outcome could be expected because slight changes in the detected sphericity and the depiction in rectangular pixels impact $d_{max}$ much more. Additionally, $d_{max}$ is defined in a way that it has to be larger than $d_{eq}$. Moreover, 'outliers' are visible at all sizes which result from spheres that either cling together or appear elliptical in one of the two camera images.



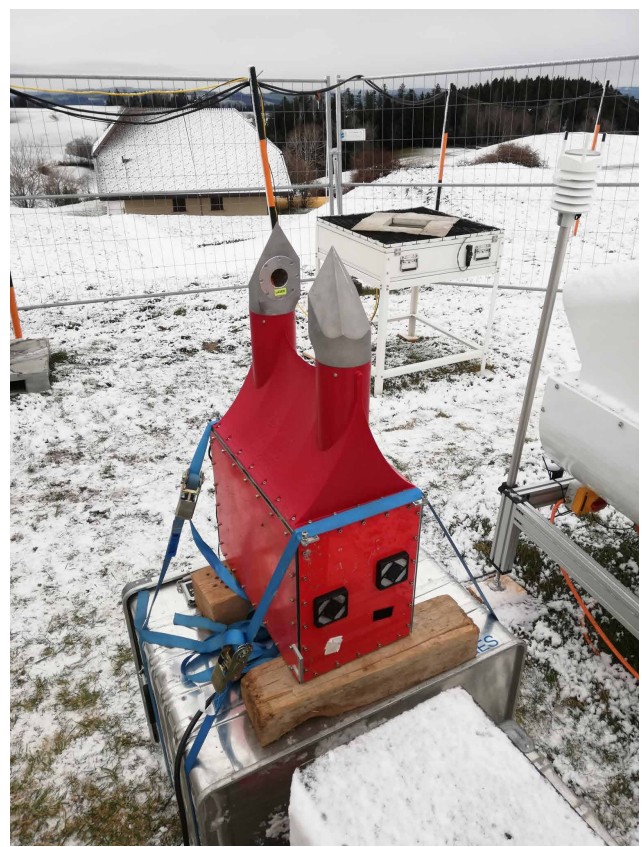

**Figure 6.** HOLIMO (red in the foreground) is operating next to the 2DVD at Rapier-Platz on 17 January 2023.

## 4.2 Detection of ice crystal shapes

On 17 January 2023 during the Cloudlab campaign in Eriswil, the holographic imager HOLIMO was operated on the ground next to the 2DVD at Rapier-Platz (Fig. 6). An occlusion front over the station led to a stratiform precipitation system, which

is visible in the time–height cross-section of the radar reflectivity (Fig. 7) with ground temperatures clearly below 0°C. Solid precipitation with different ice crystal shapes at different times was observed by eye and recorded in an educated-eye protocol. Between 14:07 and 14:11 UTC, a "large number of dendrites" was reported. For that time period, 2DVD and HOLIMO data are analysed below.

The 2DVD $O$ and $\nu$ were compared to $d_{\max}$ in order to identify clusters which represent different particle types (Fig. 8a

and b). The detected particles can be separated into $O \leq 0.6$, meaning significantly smaller height than width, and $O > 0.6$. If $O$ exceeded 2, shapes were often found to be unrealistic due to long vertical edges. This phenomenon was also seen, for example, by Grazioli et al. (2014) who suggested that particles linger in the virtual measurement area for an unusual long time span. If $O \leq 0.6$, hydrometeors were also found to fall significantly slower. Characteristic examples from the two clusters show that particles with $O \leq 0.6$ can be assumed to represent columnar crystals (Fig. 8d). For particles with $O > 0.6$, a clear shape

 

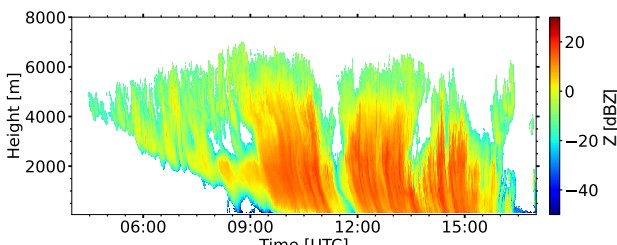

**Figure 7.** 94 GHz RPG_LACROS radar reflectivity factor on 17 January 2023 above Rapier-Platz near Eriswil, Switzerland. An offset of 20 dB compared to the 35 GHz Mira cloud radar which only operated partly due to malfunction was added.

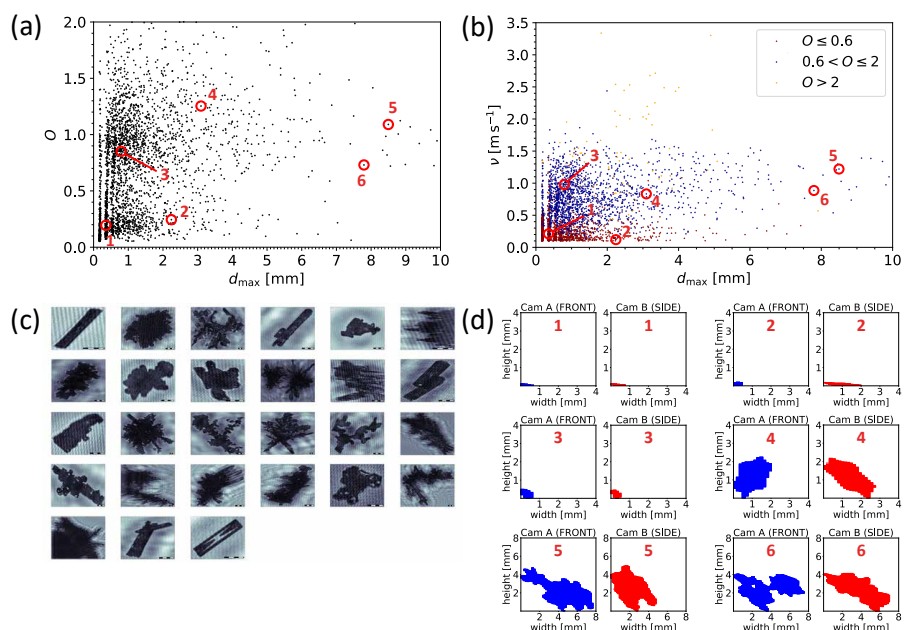

**Figure 8.** 2DVD measurements on 17 January 2023 from 14:07 UTC to 14:11 UTC including (a) $O$ versus $d_{max}$, (b) $\nu$ versus $d_{max}$, (c) all detected particles by HOLIMO during that time period, and (d) single characteristic hydrometeors. The blue numbers indicate the same hydrometeor in (a), (b), and (d). Particles 5 and 6 in (d) have a different scale than the other particles. The scale in the lower right corner in (c) corresponds to 100 $\mu$m with 25 $\mu$m subdivision.

cannot be assigned to the smaller particles as they appear only as a few pixels in the binary images. But if diameters were larger than approximately 1.5 mm, many plate-like or columnar crystals and also aggregated particles could be identified.

HOLIMO images shown in Fig. 8c prove the presence of columns and aggregates. Additionally, a certain degree of riming was observed on some crystals. Some particles should be classified as irregularly shaped. However, the major disagreement with the 2DVD findings is that dendrites and plate-like crystals could not be observed. At this point, it should be mentioned again that the probability that HOLIMO fully detects a particle decreases with increasing particle size.





### 4.3 Detection of cloud-seeding-induced precipitation particles

For a better understanding of cloud-microphysical processes, seeding experiments were conducted in the frame of Cloudlab and PolarCAP. During such experiments, a drone carrying seeding flares flies into a MPC. When the flares are combusted, silver iodide particles are generated and may initiate ice crystal nucleation (Henneberger et al., 2023). At temperatures already

slightly below freezing, the ice particles then grow and start to precipitate. If the seeding location in the cloud was chosen accordingly, HOLIMO attached to the helikite (Ramelli et al., 2020) can observe the formed cloud-ice crystals, remote-sensing instruments can monitor potential changes in cloud properties, and in-situ instruments on the ground might even detect precipitation particles, if they reached the ground at the measurement site. The latter was the case on 25 January 2023 during two seeding experiments starting at 18:55 UTC and 19:48 UTC. The RPG_LACROS reflectivity ($Z$) and linear depolarisation ratio

(LDR) from 16:00 UTC to 21:00 UTC is shown in Fig. 9a and b. The presence of a typical low-level stratiform Bise cloud offered excellent synoptic conditions for seeding experiments. The cloud consisted mainly of liquid water droplets, but from time to time, ice crystals were formed at the cloud top which then grew on the expense of liquid water and caused an increase of radar reflectivity. The two seeding experiments are reflected in both increased reflectivity and LDR for several minutes.

During the afternoon and evening on 25 January 2023, the 2DVD $N$ shows two sudden increases (Fig. 9c), each only for

two or three minutes, around 15 minutes after the start of each seeding event. The times of increased $N$ are identical with those of increased reflectivity and LDR, which suggests that the formation of according particles detected by the 2DVD was initiated by the aerosols of the burning flares.

To analyse the precipitation induced by the seeding events, three time periods are investigated with a focus on detected particle shapes. The first period was chosen from 18:00 UTC to 19:00 UTC for a background reference to get information

about hydrometeors precipitating without seeding impact. The second period from 19:10 UTC to 19:13 UTC and the third period from 20:00 UTC to 20:03 UTC cover the peaks in $R$ after each seeding event. During the first (60-minutes) period, the 2DVD detected only 51 particles.

During the three-minute periods two and three, 235 and 176 particles were detected, respectively, which indicates the seeding impact on the precipitation. The relation of $\nu$ and $d_{\max}$ for each particle of all three periods is shown in Fig. 10, including 2DVD

images of characteristic hydrometeors. Most of the particles that were observed during period one before the seeding events varied in $d_{\max}$ from 0.2 mm to 0.3 mm and in $\nu$ from 0.75 m s$^{-1}$ to 1.5 m s$^{-1}$.

Characteristic hydrometeors for that period indicate columns as major particle type which were partly rimed to some extent. This accounts for periods two and three as well, which is supported by LDR values of up to around -15 dB. Further, a temperature sensor attached to the seeding drone measured -5°C during each flare combustion, which is a suitable environment for

column and needle formation (Bailey and Hallett, 2009). However, the particles were much smaller with $d_{\max}$ mostly below 0.2 mm and $\nu < 0.5$ m s$^{-1}$ which may be due to a shorter residence time in the cloud. Particles observed in the first period could be larger because they might have stayed longer in the cloud due to turbulence, which also results in stronger riming and thus higher $\nu$. The 2DVD results have again revealed that from the instruments measurements, one can distinguish between different ice crystal shapes and thus, the 2DVD can contribute to the investigation of aerosol-cloud-interaction if precipitating



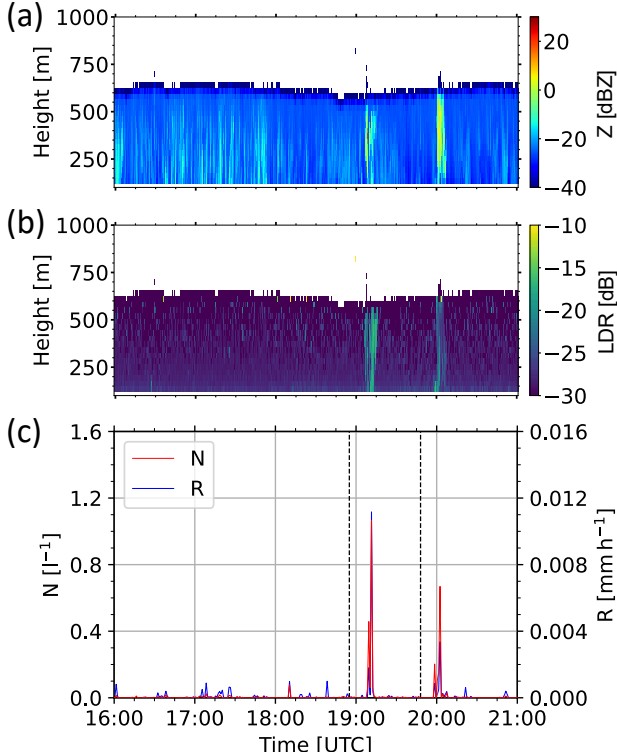

**Figure 9.** (a) 94 GHz RPG_LACROS radar reflectivity factor, (b) LDR, and (c) 2DVD $N$ and $R$ from 16 UTC to 21 UTC on 25 January 2023. The height refers to the distance above ground level. The vertical dashed lines indicate artificial cloud seeding events. Changing cloud and precipitation properties were observed some minutes after seeding.

hydrometeors reach the ground. The ground-measurement of individual particles produced by the seeding also constitute an important constrain for the interpretation of the seeding studies, as they provide information about the growth rate of the ice crystals and the evolution of ice crystal size distribution in general (e.g. Ramelli et al., 2024).

### 4.4   2DVD application to a microphysical retrieval: the 10 November 2019 MOSAiC case study

The lidar-radar retrieval LIRAS-ice, introduced by Bühl et al. (2019), allows the height-resolved determination of ice crystal
number concentration from ice or mixed-phase clouds without the necessity of in-situ measurements. Ansmann et al. (2025) and Jimenez et al. (2025) applied the method in the frame of MOSAiC case studies in order to investigate smoke-cirrus interaction and the longevity of Arctic MPCs, respectively. In this section, for the first time, LIRAS-ice is evaluated against the 2DVD as ground-based precipitation sensor by means of comparing LIRAS-ice ICNC and mean particle diameter with 2DVD $N$ and mean $d_{\max}$, respectively.

To find comparable data, several meteorological requirements which are described in Sect. 3.3 need to be fulfilled. On 10 November 2019 between 11 and 15 UTC, a precipitation system passed over the research vessel *Polarstern* offering ideal



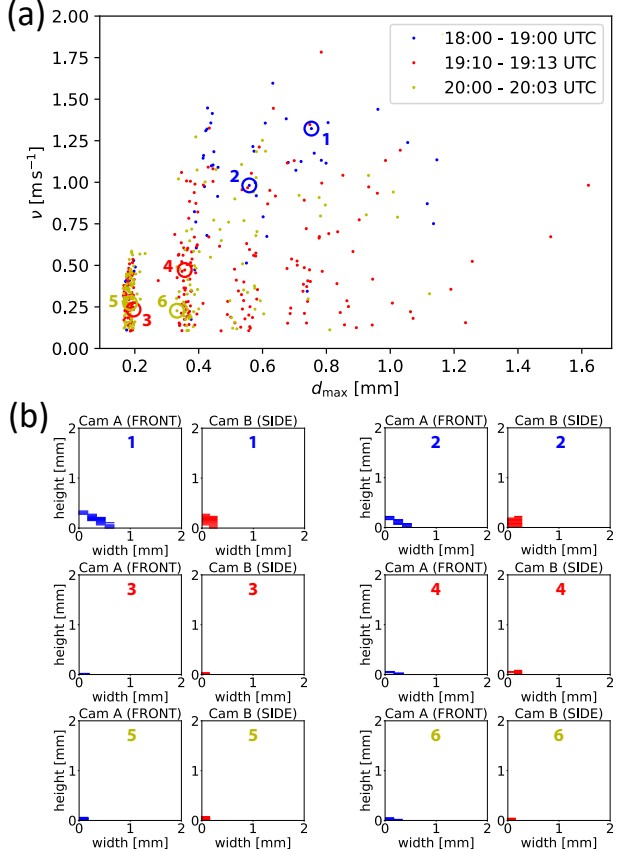

**Figure 10.** $\nu$ vs. $d_{\max}$ for different time periods on 25 January 2023.

meteorological conditions. Radar reflectivity ($Z$) values of up to 10 dBZ indicate that moderate precipitation took place during the whole time period (Fig. 11a). Within 2000 m above the research vessel, temperatures ranged between –24°C and –30°C and dropped to –40°C at the top of the highest-reaching cloud layers at around 4000 m. During almost the whole period, the 2DVD detected snowfall on the ground. Two-dimensional cross-sections of characteristic hydrometeors are additionally shown in Fig. 11a. Those particles were arbitrarily chosen from all detected hydrometeors with $d_{\max} > 1\,\mathrm{mm}$ and appear as linear branches merging in the centre, allowing an interpretation as bullet rosettes.

Figure 11b shows 30 s mean values of LIRAS-ice ICNC and 2DVD $N$. 30 s mean maximum diameters from both methods are presented in Figure 11c. All LIRAS-ice results represent the height average from the lowest height bin to 400 m for each time point in order to mitigate measurement noise. The ICNC derived under the assumption of bullet rosettes agrees with the 2DVD $N$ within one order of magnitude for most of the time. Considering that the precipitation properties can still change between ground and a few hundred meters above, the two variables generally agree well. The same accounts for the LIRAS-ice



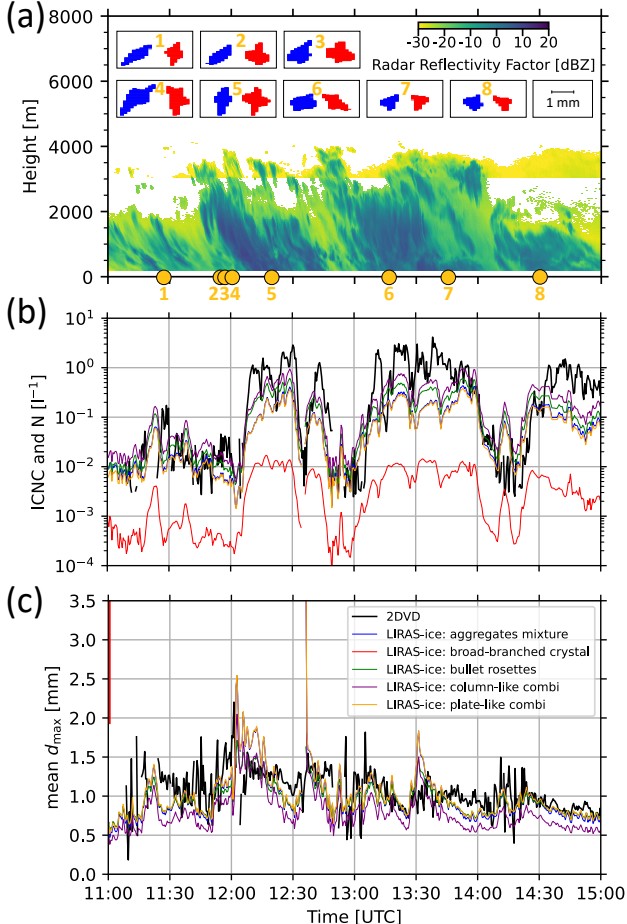

**Figure 11.** LIRAS-ice versus 2DVD results of measurements during MOSAiC in the Arctic Ocean on 10 November 2019 including different shape assumptions for LIRAS ice. The 2DVD mainly detected bullet rosettes. The figure includes (a) KAZR reflectivity factor and exemplary 2DVD particles with front view (blue) and side view (red), (b) LIRAS-ice ICNC and 2DVD $N$ time series, and (c) LIRAS-ice and 2DVD mean $d_{max}$ time series.

mean maximum diameter and $d_{max}$ from the 2DVD. The diameters show good agreement in the time series, particularly when bullet rosettes are assumed for LIRAS-ice.

Due to the time span which the particles need to fall from the analysed LIRAS-ice height range to the ground, a temporal cross-correlation between ICNC and $N$ was investigated. The highest correlation was found if the 2DVD $N$ values are shifted 60 s backwards. Further, the two-minute rolling mean was calculated for both the 2DVD and LIRAS-ice data. Rolling means of mean maximum diameter and temporally shifted $d_{max}$ were then analysed (Fig. 12) and statistically compared using Mean Absolute Error (MAE) and Root Mean Square Error (RMSE, Table 2). Assuming bullet rosettes for LIRAS-ice yielded the

lowest MAE and RMSE, with 15% and 20%, respectively, indicating moderate to good agreement with the 2DVD observations.





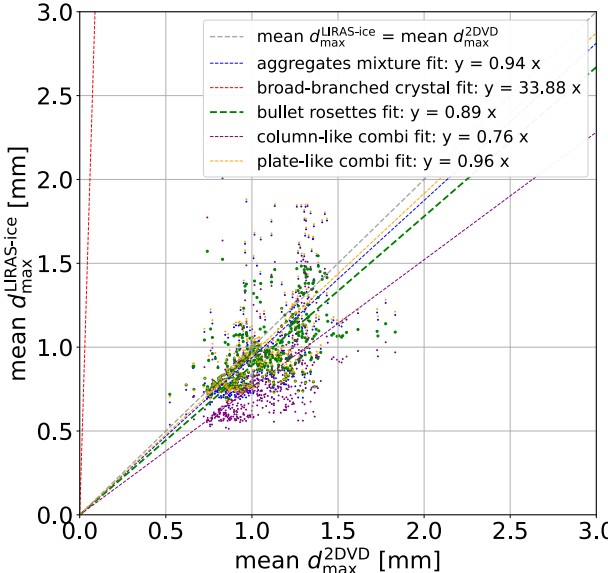

**Figure 12.** Mean maximum diameters of LIRAS-ice versus 2DVD (see Fig. 11c) including linear regression through the coordinate origin. 2DVD data were shifted by -60 s (cross-correlation) and the two-minute rolling mean was applied to both datasets.

**Table 2.** Mean Absolute Error (MAE) and Root Mean Square Error (RMSE) for 2DVD versus LIRAS-ice mean $d_{\max}$ with different assumed shapes. The average mean $d_{\max}$ of the 2DVD values is 1.062 mm and was used to calculate the relative errors.

| Dominating shape | MAE (relative) | RMSE (relative) |
|---|---|---|
| Aggregates mixture | 0.201 mm (18.91%) | 0.282 mm (26.54%) |
| Broad-branched crystals | 36.08 mm (3396.87%) | 36.34 mm (3421.44%) |
| Bullet rosettes | 0.162 mm (15.22%) | 0.215 mm (20.21%) |
| Column-like combi | 0.303 mm (28.57%) | 0.355 mm (33.43%) |
| Plate-like combi | 0.187 mm (17.62%) | 0.274 mm (25.81%) |

This is supported by a slope of 0.89 for the linear regression through the origin (Fig. 12). The assumed shapes 'aggregates mixture' and 'plate-like combi' also produced comparably small statistical errors and regression slopes close to 1. LIRAS-ice results with the other tested shapes were found to agree worse with the 2DVD observations.

## 5  Discussion

The results presented above have shown that $d_{\max}$ is a useful parameter for the 2DVD data analysis. In the following, some aspects of the findings from this study are discussed.




The application of the 2DVD $d_{\max}$ retrieval to the calibration-sphere dataset from the PolarCAP campaign revealed a slight overestimation of the sphere sizes between 0% and 10% in most cases. Some 'outliers' for which $d_{\max} \gg d_{\mathrm{eq}}$ were present for most likely two reasons. One reason is that some spheres with smaller diameters can cling together during the calibration
due to electrostatics. The second reason is that especially smaller spheres might not have fallen exactly vertically through the measurement area. This problem especially occurs for the spheres with diameters of 2 mm and smaller as they are more prone to horizontal wind impact. As a consequence, those particles might appear skewed in the 2DVD data. The effect is reflected in more systematic patterns, meaning slightly larger $d_{\mathrm{eq}}$ and significantly larger $d_{\max}$ than the actual sphere diameter (Fig. 5).

The method to determine dominant particle shapes from 2DVD observations during a certain time period described in Sect.
3.2 works mostly well. Typically, clusters representing different particle species can be identified in respective diagrams as the example presented in Sect. 4.2 shows. However, in case of particles smaller than $1 – 1.5$ mm, depending on the particle type, shapes can usually not be identified reliably. Bullet rosettes and column-like crystals could be identified if $d_{\max}$ exceeded 1 mm, but dendrites were only recognisable as such if $d_{\max}$ was larger than 1.5 to 2 mm.

During the cloud seeding experiments presented in this study, the 2DVD delivered reliable information about the ground
precipitation and allowed the determination of columnar crystals as dominating particle shape. These columnar crystals agree with the radar-based high LDR values in the cloud shortly after each seeding event took place. However, the columnar shape of the particles might have been advantageous for shape determination as other types are difficult to identify in case of such small particles.

The comparison of LIRAS-ice and 2DVD results of the MOSAiC case study showed that the different methods generally
agree. On the one hand, this agreement demonstrates the strong performance of LIRAS-ice in determining ICNC and mean maximum diameter under ideal meteorological conditions. On the other hand, the 2DVD $d_{\max}$ was needful and proved reliable for examining precipitation properties on the ground.

The LIRAS-ice height range for such a comparison was not easy to chose. Falling ice crystals can change their morphology (e.g. grow, sublimate, aggregate, disaggregate), for which reason the lowest possible height level would be advantageous.
However, both lidar and radar measurements are inaccurate at low heights for different reasons. In the lowest height range over the lidar, the outgoing radiation and the measurement volume are not fully overlapping. This so-called overlap effect determines the backscatter signal. In case of stationary radar, the data quality can decrease towards the ground for different reasons such as signal saturation near the instrument, ground clutter, the radar's blind zone, and decreasing vertical resolution. Therefore, the chosen height range of 180 m to 400 m for averaging is plausible.
Figure 11a revealed that the intensity of vertical wind shear at the lowest few hundred meters varied with time. Thus, the temporal shift of 2 minutes resulting from the cross-correlation represents the average from 11 UTC to 15 UTC. For shorter periods, different temporal shifts between LIRAS-ice and 2DVD data might lead to higher correlations which means that the statistical results for the whole period are biased by the varying vertical wind shear.



## 6 Summary and Conclusions

In this work, the particle maximum diameter $d_{max}$ was introduced as a new 2DVD parameter in order to make measurements better comparable to precipitation data by other instruments. Up to now, $d_{eq}$ was the standard 2DVD parameter for the description of the particle size but as it is related to $V$ of the particle, it can be highly biased by particle shape or alignment relative to the cameras. Hence, $d_{max}$ represents the larger of the two particle maximum extents seen from the two camera perspectives. For this study, the retrieval was applied to 2DVD data of two measurement campaigns. During the PolarCAP/Cloudlab

winter campaign 2022/2023 near Eriswil, Switzerland, 2DVD measurements were accompanied by extensive remote sensing and further in-situ observations with the ultimate goal to conduct artificial cloud seeding experiments. On the one hand, the setting allowed the comparison of 2DVD data with simultaneously taken in-situ ground measurements by the holographic imager HOLIMO. The data analysis showed that particle shapes can be well identified if diameters are larger than $1 - 1.5\,\text{mm}$, depending on the shape. On the other hand, seeding-induced ice crystals could be captured by the 2DVD which again proved

its ability to reliably contribute to precipitation-related studies. The second campaign, the MOSAiC expedition in the Central Arctic in winter 2019/2020, facilitated combined 2DVD and remote sensing observations as well. The dataset of 10 November 2019 was used to evaluate the remote-sensing-based LIRAS-ice retrieval with ground-based in-situ snowfall observations. During the investigated time period, the 2DVD mainly detected bullet rosettes. When LIRAS-ice used this particle type as input, statistical errors were indeed the lowest among all tested ice crystal shapes. The good agreement of LIRAS-ice ICNC

and mean maximum diameter with the 2DVD $N$ and mean $d_{max}$, respectively, demonstrate the applicability of the maximum diameter retrieval introduced in this study.

   The main conclusion of this study is that the new variable $d_{max}$ valuably expands 2DVD data analysis capabilities as demonstrated in the frame of different observational settings in this work. Especially the strong agreement with remote sensing observations justifies to consider the incorporation of the instrument in future studies to enhance the information about shape,

size and number of precipitation particles. Due to the 2DVD's particle typing and sizing capability, conclusions about the polarimetric properties and the valid scattering regime (such as Rayleigh or Mie) of precipitation systems can be drawn (von Terzi et al., 2025; Myagkov et al., 2025). Therefore, the instrument's potential for evaluating further remote sensing retrievals, in which for example polarimetric radar observations are included (e.g. Myagkov et al., 2016), should be highlighted in this context. For subsequent studies, we suggest to further evaluate and, if necessary, refine $d_{max}$. The calibration dataset has

shown that the relative difference of the steel spheres' actual diameters and $d_{max}$ increases as the sphere size decreases. Under different wind conditions, for example, results might differ so that calibration datasets and wind speed could potentially be used for $d_{max}$ correction. Moreover, well-defined non-spherical particles could additionally be used in the future and may improve the calibration outcome. Furthermore, $d_{max}$ should be used for automated particle typing as was done with other variables in previous studies (Grazioli et al., 2014; Gavrilov et al., 2015; Lee et al., 2015; Bernauer et al., 2016). Automated particle typing

would improve the 2DVD data analysis by enabling comprehensive remote sensing retrieval evaluation and comparison, and it would make the instrument even more attractive for further cloud and precipitation research.



*Code and data availability.* The python script which generates netCDF files containing the 2DVD particle maximum diameter can be found in Gaudek et al. (2025a). All measurement data of the 2DVD, as well as the radar data used within the PolarCAP/Cloudlab case studies are available in Gaudek et al. (2025b). The Polly lidar level 0 data used in the MOSAiC case study are in the PollyNet database (Polly, 2025),
MOSAiC cloud radar data were taken from the ARM database (ARM, 2025; ARM-MOSAiC, 2025).

*Author contributions.* TG wrote the paper and developed the $d_{\max}$ algorithm with support from AK and JB. AA and CJ were involved in the design of and discussion about the MOASiC case study. LIRAS-ice was developed and applied by JB and CJ, respectively. CF and JH conducted the HOLIMO measurements and provided the final HOLIMO data. TG, KO, and PS conducted the PolarCAP measurements. RE was responsible for the MOSAiC measurements. PS supervised this study.

*Competing interests.* The authors declare that they have no conflict of interest.

*Acknowledgements.* We would like to thank everyone who participated in the planning and execution of the Cloudlab/PolarCAP and MOSAiC campaigns, thus enabling the valuable datasets. We also thank Joanneum Research, the manufacturer of the 2DVD, for their kind support with the 2DVD data processing.

*Financial support.* Funding for this study was provided by the European Union's Horizon Europe project CleanCloud
(grant no. 101137639), by the Deutsche Forschungsgemeinschaft (DFG, German Research Foundation) within the priority programme SPP 2115 PROM via project number 408027490 (PolarCAP) and via project no. 268020496 of TRR172, and by European Research Council (ERC) under the European Union's Horizon 2020 research and innovation programme (Grant Agreement 101021272 CLOUDLAB). The MOSAiC programme was funded by the German Federal Ministry for Education and Research (BMBF) through financing the Alfred Wegener Institut Helmholtz Zentrum fürPolar und Meeresforschung (AWI)
and the Polarstern expedition PS122 under grant N-2014-H-060_Dethloff. Analysis of the MOSAiC data was funded by BMBF project SCiAMO (MOSAIC-FKZ 03F0915A).



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
