# Peer review of "Contribution of the 2DVD to the investigation of cloud microphysics during the MOSAiC and Cloudlab/PolarCAP campaigns"

_EGUsphere, 2025_

## Referee Comment (RC1)

Review of "Contribution of the 2DVD to the investigation of cloud microphysics during the MOSAiC and Cloudlab/PolarCAP campaigns"

This manuscript evaluates a 2DVD maximum diameter estimate that is computed using the larger of the two Feret diameters from the pair of 2DVD cameras. Two approaches are used to evaluate the maximum diameter estimates: direct evaluation using calibration spheres of known diameter and comparison against a lidar-radar remote sensing retrieval.

Additionally, the manuscript demonstrates the applications of the maximum diameter estimate for use in crystal habit identification, comparing the results with another in-situ instrument. Overall, I think the manuscript has merit but will need some substantial improvements prior to publication. As such, I'm recommending major revisions. I've included some general comments below followed by specific comments with corresponding line numbers. I have also included some suggested rewordings as I came across them, although I will caveat these with the note that I'm American and I assume EGUsphere leans towards British English wording.

**General Comments:**

- 1. As mentioned above, the English needs to be improved prior to publication
- 2. I found myself asking the question "why not estimate the maximum diameter of the 3D volume reconstructed following the method 2DVD uses (where each layer is assumed to be an ellipse) instead of using the Feret diameter approach?" This is not to say that I don't think the Feret diameter approach demonstrated here is without merit, just that the authors may wish to address the choice of using the Feret diameters rather than the 3D reconstruction. I wouldn't be surprised if a 3D reconstruction of snowflakes and aggregates is prone to overestimating the volume (and presumably the maximum dimension) and referencing literature discussing the issues with the 2DVD's 3D reconstruction for snowflakes may be a helpful way to support the value of using the Feret diameter approach rather than a 3D reconstruction approach.
- 3. The manuscript would benefit from a discussion of how the maximum diameter estimate will be biased by horizontal particle motion. The bias due to horizontal motion should be fairly easy to estimate if the horizontal motion of the particle is known (obviously not possible with the actual observations, unfortunately). I feel like it should be fairly straight forward to derive an equation to relate the horizontal particle motion to the resulting bias in the Feret diameter for an individual camera and such an equation may prove useful to users looking to understand the error characteristics of this new measurement.

**Specific Comments:**

Title: The title would benefit from some mention of the maximum diameter estimate since introducing and evaluating this estimate is the core goal of the paper

Line 112: Was AMF-1 deployed on *Polarstern*? If so, it may help to remind the reader of this by adding something like "aboard *Polarstern*" to the end of this sentence.

Line 123: I like that you mention the wind speeds as this was something I was wondering about. As a side note, removing "the conduction of" might improve the clarity of this sentence.

Line 159: I'm guessing "dropped" would probably be a more accurate word than "thrown" here (as well as on line 163), but I'm not familiar with the specifics of the 2DVD calibration (beyond what has been mentioned here, of course). Also, are the spheres dropped one at a time or in batches? Later in the text, it is mentioned that the spheres might stick together, which makes me think they might be dropped in batches. It's probably worth clarifying this point in this section.

Lines 232 – 233: Given the importance of the Feret diameter to this paper, I think it may help the readers to contrast the Feret diameter with a traditional diameter (I had to search online to understand the difference). Specifically, assuming my understanding of Feret diameter is correct, that the Feret diameter is the diameter of a 3D object from a single point a view.

Section 3.3: If you can find the information in the literature, it might be helpful to briefly mention what data was used to develop the LIRAS-ice retrieval. Given that this manuscript is introducing the Feret-diameter-based maximum diameter estimate, the typical concern about using the training data to validate an algorithm isn't an issue. That said, explicitly stating what instrument was used as the ground truth for the LIRAS-ice algorithm development would still be nice. Additionally, it would be helpful mention the vertical and temporal sampling intervals for the LIRAS-ice data in this section (alternatively, the LIRAS-ice data set could potentially be added to Table 1 by changing the "Instrument" column heading to "Data Source" or something similar). Finally, please also mention which instruments are used here to provide the input data for the LIRAS-ice retrieval (my guess would be the PollyXT lidar and the 94 GHz radar).

Line 310: Regarding "spheres that either cling together", this is where I was unsure how the 2DVD calibration is performed. If the spheres are dropped one at a time, then spheres clinging together either shouldn't be an issue or would be a bad calibration that should be excluded from the calibration data. Please clarify this is section 2.3.2.

Line 324: Regarding "assumed to represent columnar crystals", the wording here makes it unclear how thorough the authors were in determining what type of particles correspond to  $O \le 0.6$ . It might be helpful to expand on this slightly, even if just by mentioning that the authors visually inspected a sizable subset of the particles during this period and found columnar crystals to be the dominant particle type. The current wording also suggests an absoluteness that should probably be softened given that (presumably) the authors did not exhaustively examine all the particles with  $O \le 0.6$  that were observed by the 2DVD during this period. Also, is there a reason that these could not be needles?

Figure 8a: Consider adding a horizontal reference line indicating O = 0.6 (I leave this up to author preference)

Lines 341 – 343: I feel like there's some ambiguity in the sentence that is leaving space for incorrect interpretations. In my experience, ice clouds will generally have a lower reflectivity than liquid water clouds and snow will generally have a much lower reflectivity than liquid precipitation. I suspect what the authors are trying to say here is that the reflectivity increase was due to the ice crystals at the top of the cloud growing to precipitation-sized ice particles at the expense of liquid cloud droplets. It might also help with clarity to specify that these were supercooled liquid water droplets (assuming that was the case).

Line 378: It would be helpful to mention how these temperature measurements were obtained (e.g., balloon soundings?). Also, I suggest replacing "Within" with "Below".

Line 384: Please mention the height of the lowest height bin used in the average.

Lines 384 – 385: In the discussion, the authors mention that vertical wind shear was a concern. If this is the case, have you considered using only the lowest few height bins in the average and averaging over time instead to reduce the noise?

Lines 391 – 392 regarding the 60 s lag correlation working best: Based on Fig 10a, the typical fall speeds of the ice particles were around 1 m/s at the surface while, according to the discussion section, the LIRAS-ice data is averaged between 180 and 400 m heights. In a theoretical situation with zero vertical wind shear, my expectation would be that the shortest physically meaningful lag time would need to be 180 seconds (and somewhere around 290 seconds would probably be more reasonable) as that is how long it takes the particles to reach the 2DVD after falling from the lowest point in the LIRAS-ice data used in the correlation. Adding in wind shear complicates this, of course. A brief search suggests there were regular radiosonde launches from *Polarstern*, so it should be possible to for the authors to examine the wind shear vectors (in a du/dz and dv/dz sense) and compare them to the ship's direction of travel to determine how the wind shear profile would affect the

particles as they fall (possibly by assuming a 1 m/s fall speed and integrating the wind shear vertically). Even if the authors decide not to go so far as to estimate the expected lag based on shear and fall speed, I feel the result warrants further discussion that includes evidence to support the relatively short lag time.

Lines 405 – 407: Unless the calibration occurred under particularly windy conditions, shouldn't the lateral movement be fairly small? I know the spheres are very small, but the distance they need to fall also looks fairly small based on Fig 2c. If you know when the calibration occurred, it might be worth checking the corresponding wind data. Also, the authors should mention the errors introduced when a small sphere is pixelized (I'm pretty sure this was mentioned earlier in the paper, but it is worth repeating here)

Lines 419 – 422: This paragraph strikes me as circular reasoning: the 2DVD Feret-based maximum diameter estimate is good because it matches LIRAS-ice and the LIRAS-ice data is good because it matches the 2DVD Feret-based maximum diameter estimate. While I'd agree that the agreement of the two gives greater credibility to the individual data sets, presumably the LIRAS-ice data has been validated elsewhere in the literature and the agreement of the two is simply showing that the 2DVD Feret-based maximum diameter estimate is reasonable.

Lines 423 – 429: The authors might also want to mention the benefits of averaging in reducing noise, which was listed as the main motivation when talking about the averaging in the main body of the manuscript.

Line 427 – 428 regarding "decreasing vertical resolution": If this is a vertically pointing radar, I don't think the vertical resolution would change with range. The across-beam resolution would change (i.e., horizontal resolution for a vertically pointing radar), perhaps this is what the authors meant?

**Technical Corrections/Suggestions:**

Line 15: "instruments" should be "instrument's"

Table 1 Caption: Should "developed" be "deployed separately"? The first and second halves of the second sentence don't seem to match up with one another if the word "developed" is the correct word. Also, mention in the caption that the superscripts indicate the instrument location.

Table 1: "Resolution" isn't really an accurate heading for this column. I'm not entirely sure what the best heading would be, however. The values appear to be a mixture of gate spacing, sampling frequency, uncertainties, and limitations. Maybe "Measurement Parameters"? Not sure.

Line 130: Suggest changing "the instrument is designed advantageous" to "the instrument design is advantageous"

Line 132: Suggest replacing "would not be processed" with "will have minimal impact on the measurements."

Lines 132 – 133: Suggest replacing "that snow accumulates" with "snow accumulation" and "and gets blown into the virtual measuring area by wind gusts" with "that could otherwise interfere with measurements"

Line 137: Wouldn't the constructed particle shape be two-dimensional rather than one-dimensional?

Line 140: Suggest "there are" be added before "several criteria"

Line 140: Suggest "needs to fulfil are considered" be replaced with "must fulfil"

Line 142: Suggest changing "In case" to "In the case"

Line 185 – 186: Regarding "horizontal winds that tilt falling hydrometeors", if you are referring to the effects of horizontal motion on the reconstructed 2DVD image, "skew" is probably a better term than "tilt" just to clarify that you are not referring to the physical snowflake being rotated.

Line 187: I think this is a stray fragment that didn't get deleted during a previous edit of the manuscript.

Line 219: Suggest replacing "2 mm of a well defined" with "2 mm within a well defined"

Line 232 – 233: Suggest adding "the" before "Feret diameter"

Line 241: Suggest removing "by" from "width is usually by many times larger"

Line 252: Suggest changing "height and particle" to "height while particle" and adding "than this" after "typically higher"

Line 265 (and elsewhere): I suspect that "dominant" is the word the authors are looking for rather than "dominating"

Line 280: Suggest replacing "resembles best" with "best resembles"

Line 285: Suggest removing "further" (or replacing it with "hereafter")

Lines 286 – 300: These paragraphs feel like they belong in a case selection section rather than as part of the current section

Line 292: Suggest removing "yet"

Line 299: Suggest changing "met the requirements most" to "best met the requirements"

Line 306: Suggest removing "well" from "can well be identified"

Line 346: Suggest replacing "according" with "corresponding"

Lines 348 – 367: I feel like these paragraphs are broken up in the wrong locations. Perhaps including the first sentence of the paragraph starting on Line 353 with the previous paragraph and moving the rest of this paragraph (i.e., "The relation of..." and onwards) to the start of the following paragraph?

Line 353: Suggest removing "the three-minute" and then adding "(three minutes each)" after "periods two and three". Currently it sounds like there's an extra three-minute period before the second and third periods.

Line 357: Suggest adding "the" before "major"

Line 366: Replace "constrain" with "constraint"

Line 373: Add "a" before "ground-based"

Line 380: Replace "on the ground" with "at the surface"

Figure 11: Suggest using the same color bar for both Fig 7 and Fig 11

Figure 11 caption: I suspect the word "example" is a better fit for the authors intent than "exemplary"

Line 390: Suggest replacing "which" with "required for" and removing "need"

Line 421: Suggest replacing "needful" with "useful"

Line 429: This sentence is a bit awkward and could use rewording

Line 431: This sentence lists a 2 minute lag, but the main body of the text said a 60 s lag was used.

Line 437: Should the letter "V" instead be a lower-case nu?

Lines 453 – 455: This sentence is a bit awkward and could use rewording

---

## Author Comment (AC1)

Dear Reviewer #1,

We thank you very much for carefully reading our manuscript and for your numerous comments and suggestions! In this reply letter, the comments of Reviewer #1 are given in **black** and our answers are written in **green**. The given line numbers refer to the revised version of the manuscript (without markup).

**General comments**

1. As mentioned above, the English needs to be improved prior to publication

> Thank you again at this point for the many comments and suggestions concerning the linguistics of the draft. We implemented all of your suggestions and corrected a few more mistakes that we or Reviewer #2 found.

2. I found myself asking the question "why not estimate the maximum diameter of the 3D volume reconstructed following the method 2DVD uses (where each layer is assumed to be an ellipse) instead of using the Feret diameter approach?" This is not to say that I don't think the Feret diameter approach demonstrated here is without merit, just that the authors may wish to address the choice of using the Feret diameters rather than the 3D reconstruction. I wouldn't be surprised if a 3D reconstruction of snowflakes and aggregates is prone to overestimating the volume (and presumably the maximum dimension) and referencing literature discussing the issues with the 2DVD's 3D reconstruction for snowflakes may be a helpful way to support the value of using the Feret diameter approach rather than a 3D reconstruction approach.

> The 3D reconstruction based on stacking the ellipses on each other is a good approach for liquid droplets. However, in case of ice crystals, the volume is, as you supposed, largely overestimated. This is shown, for example, by the studies from Brandes et al. (2007) or Zhang et al. (2021) who experimentally investigated the 2DVD bulk density of ice crystals and found strong dependencies on the particle size, relative humidity, and further variables. Bulk densities down to 0.1 g/cm$^3$ were found under certain conditions. This is roughly 1/10 of the density of ice which means that those 2DVD particle volumes are differing up to one order of magnitude from the true particle volume. As Reviewer #2 pointed out that our precipitation rate in Fig. 9 is affected by that, we implemented the bulk density — size relation found by Zhang et al. (2021) into our precipitation rate calculation (Fig. 9 and Eq. 4). Following your suggestion, we added some sentences at the beginning of Section 3.1 in which we explain why we refrained from a 3D reconstruction (lines 238-242).

3. The manuscript would benefit from a discussion of how the maximum diameter estimate will be biased by horizontal particle motion. The bias due to horizontal motion should be fairly easy to estimate if the horizontal motion of the particle is known (obviously not possible with the actual observations, unfortunately). I feel like it should be fairly straight forward to derive an equation to relate the horizontal

particle motion to the resulting bias in the Feret diameter for an individual camera and such an equation may prove useful to users looking to understand the error characteristics of this new measurement.

> —> Indeed, the maximum diameter retrieval would strongly benefit from such a correction. Helms et al. (2022) have applied an unskewing algorithm to 2DVD data and found a slightly better agreement to the simultaneously acquired PIP (Precipitation Imaging Package) data. Their method works in a way that the upper and lower most detected pixel of the 2DVD image should be vertically aligned. To our opinion, this makes much sense in case of droplets. In case of ice crystals, however, uncertainties should remain high. Helms et al. (2022) accordingly also conclude in their "Conclusions" Section, that "the corrected bounding-box-width measurements are still prone to error due to the motion skewing effects" after the application of the unskewing algorithm. For this reason, we decided to not deal with skewness correction at this point, while we added the potential of a skewness correction to the Conclusions section in line 496.

**Specific comments**

Title: The title would benefit from some mention of the maximum diameter estimate since introducing and evaluating this estimate is the core goal of the paper

> —> We also thought about including the maximum diameter in the title. However, as the content also heavily deals with cloud microphysical studies in which also remote sensing instruments as well as HOLIMO are playing a crucial role, we would like to stay with the current title.

Line 112: Was AMF-1 deployed on Polarstern? If so, it may help to remind the reader of this by adding something like "aboard Polarstern" to the end of this sentence.

> —> Yes, this paragraph is only about MOSAiC and Polarstern. Nevertheless, we added "aboard Polarstern" for clarification, as suggested. (line 117)

Line 123: I like that you mention the wind speeds as this was something I was wondering about. As a side note, removing "the conduction of" might improve the clarity of this sentence.

> —> "the conduction of" is now removed.

Line 159: I'm guessing "dropped" would probably be a more accurate word than "thrown" here (as well as on line 163), but I'm not familiar with the specifics of the 2DVD calibration (beyond what has been mentioned here, of course). Also, are the spheres dropped one at a time or in batches? Later in the text, it is mentioned that the spheres might stick together, which makes me think they might be dropped in batches. It's probably worth clarifying this point in this section.

> —> "thrown" was replaced by "dropped" (line 163) and by "dropped by hand" (line 167).

Lines 232 – 233: Given the importance of the Feret diameter to this paper, I think it may help the readers to contrast the Feret diameter with a traditional diameter (I had to search online to

understand the difference). Specifically, assuming my understanding of Feret diameter is correct, that the Feret diameter is the diameter of a 3D object from a single point a view.

—> Thanks for this important hint. We decided to include the circumscribing sphere diameter (dc) in this study. To summarize the results on that: We found that the maximum Feret diameter is very similar to dc. We included dc in Fig. 5 which is showing the similarity of dmax and dc for the calibration sphere data set. Our motivation for introducing the maximum Feret diameter was that, from a physical perspective, the definition of a maximum diameter should be the distance of its two outermost points. For that reason, we think that it is sufficient to briefly show the similarity between dmax and dc and to proceed with dmax in the paper. We also briefly investigated the difference of dmax and dc for both case studies and found that the two diameter types are almost identical for the snow measurements as well (see Fig. R1).

[Figure]

Figure R1: 2DVD dmax versus dc (a) on 17 January 2023 during PolarCAP/Cloudlab in Eriswil, Switzerland and (b) on 10 November 2019 from 11 to 15 UTC during MOSAiC.

Section 3.3: If you can find the information in the literature, it might be helpful to briefly mention what data was used to develop the LIRAS-ice retrieval. Given that this manuscript is introducing the Feret-diameter-based maximum diameter estimate, the typical concern about using the training data to validate an algorithm isn't an issue. That said, explicitly stating what instrument was used as the ground truth for the LIRAS-ice algorithm development would still be nice. Additionally, it would be helpful mention the vertical and temporal sampling intervals for the LIRAS-ice data in this section (alternatively, the LIRAS-ice data set could potentially be added to Table 1 by changing the "Instrument" column heading to "Data Source" or something similar). Finally, please also mention which instruments are used here to provide the input data for the LIRAS-ice retrieval (my guess would be the PollyXT lidar and the 94 GHz radar).

—> Thank you for pointing out that this Section needed some additional explanations. To clarify which instruments were used for the development of LIRAS-ice, the following

sentence was added in lines 294 to 295: "For the development of their retrieval, Bühl et al. (2019) used data of a PollyXT Raman lidar, a 35GHz cloud radar, and a radar wind profiler (RWP), albeit the usage of the RWP is optional."

—> For the development of LIRAS-ice, clouds were investigated which did not precipitate down to the ground. Therefore, no ground-based in-situ instrument was used. This was also a particular motivation for us to compare the 2DVD data to the LIRAS-ice results.

—> To inform about the instruments and the sampling intervals in this study, the following sentences were added in lines 314 to 317: "For this study, the Raman lidar PollyXT and of the KAZR cloud radar which were operated during the MOSAiC campaign provide the observational basis (see Table 1). The vertical and temporal sampling intervals of the retrieved variables equal those of the processed cloud radar data which are approximately 30 m and 30 s, respectively."

Line 310: Regarding "spheres that either cling together", this is where I was unsure how the 2DVD calibration is performed. If the spheres are dropped one at a time, then spheres clinging together either shouldn't be an issue or would be a bad calibration that should be excluded from the calibration data. Please clarify this is section 2.3.2.

—> The calibration was done again on 11 November 2025 under calm conditions and with particular care. This improved the results a lot by means of a much smaller distribution for each single sphere size and also the number of outliers got reduced a lot. Nevertheless, the smallest sphere sometimes stick to other spheres due to electrostatic charge or magnetism and can easily be overseen. Following your suggestion, they were manually filtered from the new dataset.

Line 324: Regarding "assumed to represent columnar crystals", the wording here makes it unclear how thorough the authors were in determining what type of particles correspond to O <= 0.6. It might be helpful to expand on this slightly, even if just by mentioning that the authors visually inspected a sizable subset of the particles during this period and found columnar crystals to be the dominant particle type. The current wording also suggests an absoluteness that should probably be softened given that (presumably) the authors did not exhaustively examine all the particles with O<=0.6 that were observed by the 2DVD during this period. Also, is there a reason that these could not be needles?

—> We changed this sentence to "Characteristic examples from the two clusters show that many particles with O ≤ 0.6 have vertical extents of only a few pixels while reaching horizontal lengths of up to several millimetres (Fig. 8d). Therefore, their majority can be assumed to represent columnar crystals or needles." (lines 351-353).

Figure 8a: Consider adding a horizontal reference line indicating O = 0.6 (I leave this up to author preference)

—> We added the horizontal dashed line.

Lines 341 – 343: I feel like there's some ambiguity in the sentence that is leaving space for incorrect interpretations. In my experience, ice clouds will generally have a lower reflectivity than

liquid water clouds and snow will generally have a much lower reflectivity than liquid precipitation. I suspect what the authors are trying to say here is that the reflectivity increase was due to the ice crystals at the top of the cloud growing to precipitation-sized ice particles at the expense of liquid cloud droplets. It might also help with clarity to specify that these were supercooled liquid water droplets (assuming that was the case).

> —> This is true. This one and the following sentence were modified so that the statement became more clear.

Line 378: It would be helpful to mention how these temperature measurements were obtained (e.g., balloon soundings?). Also, I suggest replacing "Within" with "Below".

> —> It is now mentioned that radiosonde data were used and a citation (Maturilli et al., 2021) was added. "Within" was replaced with "Below". (lines 410-411)

Line 384: Please mention the height of the lowest height bin used in the average.

> —> "starting at around 180 m," was inserted (lines 416-417)

Lines 384 – 385: In the discussion, the authors mention that vertical wind shear was a concern. If this is the case, have you considered using only the lowest few height bins in the average and averaging over time instead to reduce the noise?

> —> Yes, different options regarding the height bins were tested. One height bin is around 30m deep. So our range from around 180m to 400m equals the lowest seven height bins. We found that using only one height bin was disadvantageous because data gaps were too numerous and, especially in the case of the lowest height, bins with unrealistic results ('outliers') occurred too often. Further, we found that averaging to 300m, 400m, or 500m did not change the results much so that we chose 400m.

Lines 391 – 392 regarding the 60 s lag correlation working best: Based on Fig 10a, the typical fall speeds of the ice particles were around 1 m/s at the surface while, according to the discussion section, the LIRAS-ice data is averaged between 180 and 400 m heights. In a theoretical situation with zero vertical wind shear, my expectation would be that the shortest physically meaningful lag time would need to be 180 seconds (and somewhere around 290 seconds would probably be more reasonable) as that is how long it takes the particles to reach the 2DVD after falling from the lowest point in the LIRAS-ice data used in the correlation. Adding in wind shear complicates this, of course. A brief search suggests there were regular radiosonde launches from Polarstern, so it should be possible to for the authors to examine the wind shear vectors (in a du/dz and dv/dz sense) and compare them to the ship's direction of travel to determine how the wind shear profile would affect the particles as they fall (possibly by assuming a 1 m/s fall speed and integrating the wind shear vertically). Even if the authors decide not to go so far as to estimate the expected lag based on shear and fall speed, I feel the result warrants further discussion that includes evidence to support the relatively short lag time.

> —> Thank you for pointing out this interesting aspect. There was a balloon launch at 10:34 UTC and another one at 16:35 UTC. The radio sonde from the 10:34 UTC launch is showing wind speeds from 2.8 m/s (at 22m) to 5.7 m/s (150 - 200m) and directions from 164° (at 10m) to 133° (at around 180m). This means that there is both a directional as well as a

velocity wind shear between the ground and the lowest LIRAS-ice height bin. In addition to that, the wind direction has changed during the investigated time period: The 16:35 UTC launch shows directions from 170° at 20m to 182° at 180m. Between the two launches, the direction changed from 133° to 182° at 180m but remained approximately constant close to the ground which is likely reflected in the different slopes of the maximum radar reflectivities in Fig. 11 a. At 11 UTC, the slope of the fall streak is significantly different from one of the fall streaks at 13 UTC, for example. We conclude that the 60s can still considered to be plausible. This is mainly because it is an average value for which the correlation along the whole investigated time period is the best. One would, for example, only need a short time period in between, where 2DVD and LIRAS-ice investigated different particles which could affect the average correlation. Moreover, the correlation of the neighbouring time shifts (30s, 90s, etc.) was not differing too much from the one for 60s. We added the sentence "This value can be considered plausible given the presence of vertical wind shear (Maturilli et al., 2021)" in line 424.

Lines 405 – 407: Unless the calibration occurred under particularly windy conditions, shouldn't the lateral movement be fairly small? I know the spheres are very small, but the distance they need to fall also looks fairly small based on Fig 2c. If you know when the calibration occurred, it might be worth checking the corresponding wind data. Also, the authors should mention the errors introduced when a small sphere is pixelized (I'm pretty sure this was mentioned earlier in the paper, but it is worth repeating here)

  —> We repeated the calibration under zero-wind conditions. Nevertheless, a few spheres appeared skewed in the data. We think that this can be explained by the fact that they were dropped by hand through the instrument. The calibration pattern from the one figure can only be used for the 10mm spheres but the others must be dropped by hand as the manufacturer does not provide a separate calibration pattern or similar. Therefore, we assume that using an adapted calibration pattern for all spheres would improve the calibration results even further. We modified this part of the discussion and present the improved results here (lines 436-441)).

Lines 419 – 422: This paragraph strikes me as circular reasoning: the 2DVD Feret-based maximum diameter estimate is good because it matches LIRAS-ice and the LIRAS-ice data is good because it matches the 2DVD Feret-based maximum diameter estimate. While I'd agree that the agreement of the two gives greater credibility to the individual data sets, presumably the LIRAS-ice data has been validated elsewhere in the literature and the agreement of the two is simply showing that the 2DVD Feret-based maximum diameter estimate is reasonable.

  —> Indeed, the LIRAS-ice retrieval has not yet been properly evaluated against any in-situ sensors. Only Ansmann et al. (2025) compared the ice crystal number concentration in cirrus clouds with the 2DVD particle number concentration on the ground and found that they agree within one order of magnitude or better. This is the reason, why we decided to perform this comparison between LIRAS-ice for the lowest height bins and 2DVD.

Lines 423 – 429: The authors might also want to mention the benefits of averaging in reducing noise, which was listed as the main motivation when talking about the averaging in the main body of the manuscript.

> —> This is an important aspect, of course. We added "considering only one height bin would result in too strong temporal fluctuation of ICNC and maximum diameter." (line 459).

Line 427 – 428 regarding "decreasing vertical resolution": If this is a vertically pointing radar, I don't think the vertical resolution would change with range. The across-beam resolution would change (i.e., horizontal resolution for a vertically pointing radar), perhaps this is what the authors meant?

> —> You are right, the vertical resolution itself does likely not change in that height range. To generalise and simplify the statement, we changed "signal saturation near the instrument, ground clutter, the radar's blind zone, and decreasing vertical resolution" to "near-field effects and beam geometry". (line 463)

**Technical Corrections / Suggestions**

Line 15: "instruments" should be "instrument's"

> —> We followed the suggestion.

Table 1 Caption: Should "developed" be "deployed separately"? The first and second halves of the second sentence don't seem to match up with one another if the word "developed" is the correct word. Also, mention in the caption that the superscripts indicate the instrument location.

> —> We followed the suggestion, changed "developed" to "deployed separately" and added the sentence "The superscripts indicate the platform of each instrument and therefore its measurement location.".

Table 1: "Resolution" isn't really an accurate heading for this column. I'm not entirely sure what the best heading would be, however. The values appear to be a mixture of gate spacing, sampling frequency, uncertainties, and limitations. Maybe "Measurement Parameters"? Not sure.

> —> We changed "Resolution" to "Specifications".

Line 130: Suggest changing "the instrument is designed advantageous" to "the instrument design is advantageous"

> —> We followed the suggestion.

Line 132: Suggest replacing "would not be processed" with "will have minimal impact on the measurements."

> —> We followed the suggestion.

Lines 132 – 133: Suggest replacing "that snow accumulates" with "snow accumulation" and "and gets blown into the virtual measuring area by wind gusts" with "that could otherwise interfere with measurements"

—> We followed the suggestion.

Line 137: Wouldn't the constructed particle shape be two-dimensional rather than one-dimensional?

—> This is of course true, we changed "one" to "two".

Line 140: Suggest "there are" be added before "several criteria"

—> We followed the suggestion.

Line 140: Suggest "needs to fulfil are considered" be replaced with "must fulfil"

—> We followed the suggestion.

Line 142: Suggest changing "In case" to "In the case"

—> We followed the suggestion.

Line 185 – 186: Regarding "horizontal winds that tilt falling hydrometeors", if you are referring to the effects of horizontal motion on the reconstructed 2DVD image, "skew" is probably a better term than "tilt" just to clarify that you are not referring to the physical snowflake being rotated.

—> We followed the suggestion and reworded the sentence to "However, for solid particles, the applied method to calculate $O$ can be highly error-prone, especially at significant horizontal winds which may let falling hydrometeors appear skewed in the camera images." (lines 188-190).

Line 187: I think this is a stray fragment that didn't get deleted during a previous edit of the manuscript.

—> This was shifted to the caption of Fig. 4 and the code is now published on Zenodo.

Line 219: Suggest replacing "2 mm of a well defined" with "2 mm within a well defined"

—> We followed the suggestion.

Line 232 – 233: Suggest adding "the" before "Feret diameter"

—> We followed the suggestion.

Line 241: Suggest removing "by" from "width is usually by many times larger"

—> We followed the suggestion.

Line 252: Suggest changing "height and particle" to "height while particle" and adding "than this" after "typically higher"

—> We followed the suggestion.

Line 265 (and elsewhere): I suspect that "dominant" is the word the authors are looking for rather than "dominating"

> —> We replaced "dominating" with "dominant" here and at nine further positions in the manuscript.

Line 280: Suggest replacing "resembles best" with "best resembles"

> —> We followed the suggestion.

Line 285: Suggest removing "further" (or replacing it with "hereafter")

> —> We followed the suggestion and replaced it with "hereafter".

Lines 286 – 300: These paragraphs feel like they belong in a case selection section rather than as part of the current section

> —> Due to a comment of Reviewer #2, this Section was revised. However, as we systematically applied this method throughout all case studies, we would like to keep it in the "Methods" Section. To formulate it more as a method, we changed the Section name from "Dominating particle shapes in 2DVD data" to "Identification of dominant particle shapes in 2DVD data".

Line 292: Suggest removing "yet"

> —> We followed the suggestion.

Line 299: Suggest changing "met the requirements most" to "best met the requirements"

> —> We followed the suggestion.

Line 306: Suggest removing "well" from "can well be identified"

> —> We followed the suggestion.

Line 346: Suggest replacing "according" with "corresponding"

> —> We followed the suggestion.

Lines 348 – 367: I feel like these paragraphs are broken up in the wrong locations. Perhaps including the first sentence of the paragraph starting on Line 353 with the previous paragraph and moving the rest of this paragraph (i.e., "The relation of…" and onwards) to the start of the following paragraph?

> —> We followed the suggestion.

Line 353: Suggest removing "the three-minute" and then adding "(three minutes each)" after "periods two and three". Currently it sounds like there's an extra three-minute period before the second and third periods.

> —> We followed the suggestion.

Line 357: Suggest adding "the" before "major"

> —> We followed the suggestion.

Line 366: Replace "constrain" with "constraint"

       —> We followed the suggestion.

Line 373: Add "a" before "ground-based"

       —> We followed the suggestion.

Line 380: Replace "on the ground" with "at the surface"

       —> We followed the suggestion.

Figure 11: Suggest using the same color bar for both Fig 7 and Fig 11

       —> We followed the suggestion.

Figure 11 caption: I suspect the word "example" is a better fit for the authors intent than "exemplary"

       —> We followed the suggestion.

Line 390: Suggest replacing "which" with "required for" and removing "need"

       —> We followed the suggestion.

Line 421: Suggest replacing "needful" with "useful"

       —> We followed the suggestion.

Line 429: This sentence is a bit awkward and could use rewording

       —> The sentence was reworded to "Therefore, the choice of a height range of 180\,m to 400\,m for averaging the LIRAS-ice results is well justified." (lines 463-464).

Line 431: This sentence lists a 2 minute lag, but the main body of the text said a 60 s lag was used.

       —> 60 s is the right value, so "2 minutes" was replaced by "60 s"

Line 437: Should the letter "V" instead be a lower-case nu?

       —> Here, the volume is meant as deq is calculated from it. Some words were added for clarification.

Lines 453 – 455: This sentence is a bit awkward and could use rewording

       —> The sentence was reworded to "The strong agreement with remote sensing observations supports including the instrument in future studies to enhance information on precipitation particle shape, size and number." (lines 488-490)

---

## Author Comment (AC2)

Dear Reviewer #2,
We thank you very much for carefully reading our manuscript and for your comments and suggestions. In this reply letter, your comments are given in **black** and our answers are written in **green**. The given line numbers refer to the revised version of the manuscript (without markup).

**Major comments:**

Maximum diameter is generally one of the most ambiguously defined parameters for highly complex particles, such as ice or snow. I am of course aware of the fact that Dmax is widely used in retrievals, model parametrizations, and in-situ observations. But I am missing in the paper a discussion that Dmax has been defined in the past in very different ways. For example, most airborne in-situ probes derive Dmax as the size of the circumscribing sphere or spheroid.

—> We thank Reviewer #2 for the comment regarding our selection of dmax. It convinced us to review the existing 2DVD-based dmax retrievals, to determine and mention the dmax approach in the remote-sening-based retrieval LIRAS-ice, and, most importantly, to conduct a comparative study between the maximum Feret diameter and the circumscribing sphere diameter. The revision work that was put into this topic is elaborated on in the following comments of this reply letter, as well as in the reply letter to Reviewer #1, who raised similar points. In the introduction, the following sentences were added: "However, the particle size has been defined in the past in very different ways. Alone in 2DVD studies, various definitions of the diameter such as equivalent sphere diameter (e.g. Lee et al., 2015), maximum diameter ('maximum value of width and height seen in both cameras'; Bernauer et al., 2015), or area diameter (defined by the area with smallest circumscribed ellipse; Huang et al., 2015) were used. From a physical perspective, the maximum dimension is the distance between the two outermost points of the particle." (lines 65-69)

You decided in your manuscript to use the Feret diameter instead. I would like to see a comparison of your "Feret-Dmax" to more common methods, for example, the maximum size of a circumscribing ellipse.

—> We decided to introduce dmax describing the maximum dimension between the outermost points of a particle (from a certain perspective) because we see this as the physical maximum extent. A circumscribing sphere diameter, on the other hand, would for example exceed dmax in case of an equilateral triangle. Nevertheless, we now introduced the circumscribing sphere diameter dc and compared it to dmax in Fig. 5. It was found that dmax and dc are indeed nearly identical in case of spheres. We also compared dmax against dc for both case studies and found that dmax and dc are nearly identical (Fig. R1).

[Figure]

Figure R1: 2DVD dmax versus dc (a) on 17 January 2023 during PolarCAP/Cloudlab in Eriswil, Switzerland and (b) on 10 November 2019 from 11 to 15 UTC during MOSAiC.

You also don't comment on how Dmax is defined in the remote sensing retrievals that you are using in the intercomparsions later. In the worst case, discrepancies in your comparison of the retrieval results and the 2DVD might at least be partly due to different definitions of Dmax. I think this aspect should be discussed much more.

—> Thank you for pointing out this important aspect. In LIRAS-ice, the maximum diameter is defined as diameter of the smallest enclosing sphere. We added the sentence "The LIRAS-ice-retrieved maximum diameter is defined as the maximum distance between two points in the ice crystal from the zenith-pointing perspective and can therefore be directly compared to dmax from the 2DVD." in lines 313 to 314.

I have some problems with Fig. 5 and the discussion of it in the text: You have steel spheres with well-defined diameters as calibration objects. Wouldn't it make sense to produce two sub-plots with D(sphere) vs Deq and a second one with D(sphere) vs. Dmax? You say in L. 307 that "Deq agrees better than Dmax with the true diameter". But how can the reader judge that if you don't show the true diameter? With two sub-plots you could also plot the data as box-and-whisker plots that can much better illustrate the distribution of points (in your current Fig. 5 it is hard to estimate how many particles are clustering together). You write in L. 307 that Dmax overestimates the true diameter by 0-10% for diameters larger than 2mm. If I look at Deq=6mm or 8mm I can see several points being 2-3mm in Dmax away from the 1:1-line (those points exceed your 10% a lot). Why does Fig. 5 show a general (systematic) overestimation for Dmax in comparison to Deq? For perfect spheres I would expect the points to be around the 1:1 line. You say that Dmax is defined in a way that it has to be larger than Deq, but this is not clear to me in the case of spheres. Finally, are you applying a correction to the 2DVD data based on your calibration results? Maybe you wrote this in the text and I missed it.

—> We agree that the plot was not optimal for judging about the two variables. We applied major changes to that part, also following the comments from Reviewer #1. Firstly, we did the calibration again under very calm conditions meaning no wind at all. We also excluded around 15 data points from this new dataset for two reasons: 1. because either spheres were sticking together (although the calibration was done very

carefully, seems one can hardly avoid that with the designated method) and 2. because some spheres were not fully detected (very few spheres seemed to be partly detected on the edge of the measurement area although this should not happen, according to the manufacturer). Further, Fig. 5 was modified in a way that, following your suggestion, box and whisker subplots were introduced. Additionally, the new variable $D_c$ was implemented. Eventually, we did not apply a correction to the 2DVD dmax data based on the calibration results because the results of the (second) calibration were, on the one hand, fluctuating, and, on the other hand, regarded as sufficiently accurate with arithmetic means not differing by more than 3-5% from the actual sizes. Moreover, we think that an additional construction from which the spheres (of all sizes, not just the 10mm ones) can be dropped would improve the results even more, as the spheres can still get a horizontal motion component when being dropped by hand. We majorly modified the results and discussion part about the calibration (Section 4.1, lines 330-339 and Section 5, lines 436-441).

**Minor comments and typos:**
1. L 137: "a one-dimensional particle shape" Shouldn't this be two-dimensional?
    —> Yes, sure. "one" was changed to "two".

2. L 187: "For the display of single particles, an own programme was written". By whom? The authors or the manufacturer? How is the programme called and where can the reader find the programme? It is not mentioned in the Code availability section.
    —> The programme was written by us and is now available on Zenodo. This statement was shifted to the caption of Fig. 4 and is also contained now in the code availability section.

3. L 202: Ice density should be 916 kg/m³. Shouldn't the equivalent volume actually taking the much lower density of snow into account? I mean, if you assume the snowflake volume to be composed of pure ice, the precipitation rate will be huge, or?
    —> This is true. We now implemented the formula of Zhang et al. (2021) who investigated the particle bulk density in dependence on median volume diameter ($D_{eq,0}$). This procedure is now explained as well in Section "2.3.4 Precipitation properties". We recreated figure 9 with an updated precipitation rate. However, the results are only slightly differing from those with an assumed rho of 0.9. This is because the density derived from the formula of Zhang et al. had to be capped at 0.917 g/cm³ for very small particles. This value is approximately reached if particles have a $D_{eq}$ of 0.1mm and smaller which was the case during the seeding events.

4. L 235-236: How exactly are the parallel tangents drawn around the particle image? Are they supposed to be perpendicular to Dmax?
    —> No, the parallel tangents are independent from Dmax. They are drawn in a way that their distance (= minimum Feret diameter) is as small as possible. Their locations are also an output from the python feret module.

5. L 239: I would say you need a very trained eye to see a dendrite in Fig. 4
    —> From our extensive 2DVD data analysis we concluded that this particle is a dendrite as the six bulges are visible in both camera images.

6. Fig. 4 (caption): There is no thick red line in the figures.
    —> "red" was changed to "black"

7. L 267-268: I am surprised about your statement that columnar particles would fall slower than plate-like particles. For example, if you look at Fig. 6 in Mitchell, JAS, 1996 you see that columns fall way faster than plates or dendrites. Please revise.

> This is in theory true. However, Bühl (2014, dissertation; Figure 3.4) showed with the experimental-based method of Heymsfield and Westbrook (2010) that the fall velocity differences of rimed long columns, aggregates mixture, and hexagonal plates with sizes of 1 - 2 mm are smaller than 0.1 m/s. The paragraph was revised and some literature was added. It was also included that both oblateness and fall velocity should be used with care because both are bias-prone due to horizontal orientation and small-scale wind effects, respectively. (lines 276-284)

8. L 282: "This dominant crystal shape has to be presumed in advance" Which is a major weakness of the LIRAS-ice retrieval in my opinion.

> Indeed, we see it as an advantage of LIRAS-ice compared to similar retrievals such as DARDAR or Captivate that testing different shapes and investigating the impact on the ice crystal number concentration results is possible.

9. L 295: "with further assumed shapes" Do you mean different shapes?

> Exactly, this is what we did. We changed "further" to "several other".

10. 296: "require further requirements" Consider different wording.

> "suited to conduct such case studies require further requirement" was changed to "suitable for conducting such case studies requires additional criteria"

11. 324: You say that particles with O<0.6 which are shown in Fig. 8d can be assumed to be columns. This is completely unclear to me: If you observe a horizontally oriented plate or dendrite, its height will also always be smaller than its width (so small O). I would argue that you cannot reliably distinguish between columns and plates based on O.

> This is true. The reason for the statement was, that during most of our 2DVD analysis, dendrites or plate-like particles could often be identified as such because they are rarely horizontally aligned, likely due to wobbling. This would also increase O significantly. To account for the possibility of horizontally aligned particles (and thus, to soften the statement), the main body was slightly changed:
L. 347: "may" was added: "clusters which ["may"] represent different particle types".
L 351: "were also found to fall significantly slower" was changed to "were also found to fall more slowly on average"
L. 352-354: "many" was inserted: "[many] particles with O ≤ 0.6 can be assumed to represent columnar crystals" and the sentence "Due to the low $v$ it is possible that further particle types, such as small horizontally oriented dendrites, are also contained in this cluster." was added.

Even more confusing is the following: The #4 particle in Fig. 8d is identical to your example particle in Fig. 4 which you describe there as a dendrite (!). Also particles #5 or 6 in Fig. 8d look to me much more like aggregates than columns. Please clarify.

> L. 356-357: "many plate-like or columnar crystals and also aggregated particles" was changed to "dendrites, plate-like crystals, columns, and also aggregated particles"

12. Figure 7: I stumbled over the 20dB offset between the MIRA and the W-band radar. This offset is huge, so I would like to know a bit more about it. Which of the two radars had the "malfunction" and was it related to the offset? Was one of the two radars properly calibrated and

how? I assume your Dmax remote sensing retrieval depends on the absolute value of Ze, hence a discussion of this aspect is quite relevant.

—> To provide more explanation, the sentences "The offset occurred due to a malfunction of the RPG radar blower, followed by accumulating snow on the radome causing attenuation. The MIRA cloud radar, on the other hand, only measured intermittently during the relevant time period for which reason RPG radar data are shown." were added in the caption.

13. L 329-330: What is the sampling area of HOLIMO. Up to what maximum size can particles be reliably detected by HOLIMO?

—> HOLIMO's ability to detect large particles is limited by the relatively small measurement volume of 12cm$^3$. However, 2mm can be considered an approximate upper threshold (see Section 2.4, line 225).

14. Figure 9a: "Z" is defined as the 6th moment of the raindrop size distribution. As you are observing ice particles, you actually show the effective radar reflectivity factor which is usually denoted as "Ze".

—> "Z" was changed to "Ze" in Figure 9, 7, and as well as at two further positions in the main body.

15. L 372: You mention in the article several times (for example, in the abstract) that LIRAS-ice is evaluated "for the first time" against ground-based in-situ observations. I would assume that a proper evaluation of a remote sensing retrieval is presented in the same publication as the retrieval itself. I have the impression that you want to convince the reader how innovative your evaluation in this section is. I would suggest to leave that judgement to the readers or at least mention it only once.

—> "for the first time" was removed here.

16. Figure 10 (caption): Consider including a reference to Fig. 9.

—> The sentence "Time series of cloud radar profiles of Ze and LDR, as well as 2DVD N and R are shown in Fig. 9." is now included.

17. Figure 11 and discussion: I see in panel b) a systematic underestimation of at least 15% of N by LIRAS-ice for all assumed shapes. How big is the uncertainty of LIRAS-ice in general? How much would a realistic uncertainty in radar calibration of +-1.5dB affect your N and Dmax estimate? This should be discussed.

—> Thank you for pointing out this aspect. Bühl et al. (2019) are pointing out that determining the uncertainty for the ice crystal number concentration (ICNC) is very challenging and based on different assumptions. For the diameter, no specific uncertainty values are given. We added the following text in lines 317 to 320 Section 3.3): "The determination of uncertainty for the retrieved ICNC is complex and based on a number of assumptions, for example, about particle type, size distribution and mass– or velocity–size relationships. Further uncertainties of Z and E additionally propagate into the final ICNC uncertainty. According to Jimenez et al. (2025), the ICNC uncertainty, considering lidar and radar measurement errors as well as model errors, is about a factor of 2-3.". Also, we changed in the Results Section "The ICNC derived under the assumption of bullet rosettes agrees with the 2DVD N within one order of magnitude for most of the time." to "The ICNC derived under the assumption of bullet rosettes and 2DVD N agree within the given the LIRAS-ice ICNC uncertainty, i.e. a factor of 2–3

(Jimenez et al., 2025), for a substantial fraction of the time." (lines 417-419). In the Discussion Section, we added "ICNC and N values agree within the LIRAS-ice uncertainty for a substantial portion of time." (line 453).

18. L431 + L. 392: In L. 431 you mention a temporal shift of 2 minutes but in L. 392 you say the shift is only 1 minute (60s). Please clarify.
—> 60 seconds is the right value, so "two minutes" is replaced by "60 s".

19. L430: How can you estimate the vertical wind shear from Fig. 11a? If you infer this from the shape of the fall streaks, you have to assume particles with identical vertical velocity over time.
—> This is true. Following a comment of Reviewer #1, we took a look into the radiosonde wind data of this day. We firstly investigated vertical directional and velocity wind shear during the investigated period and we secondly found that the wind direction at the lowest LIRAS-ice height bin (180m) changed by 30° during that time period. We added the data source (Maturilli et al., 2021) in line 424 and in the data availability part. We changed "Figure 11a" to "Radio soundings from that day [...] (Maturilli et al., 2021)" (lines 465-466).

20. L433: "the statistical results (…) are biased by the varying vertical wind shear" Why don't you test the impact of different time shifts on your statistical results? It sounds plausible but it would be more scientific to show it.
—> In fact, the performed cross-correlation worked in a way that every possible time shift for the whole period was tested. The time shift which led to the highest correlation (which is -60 seconds) was then used for the calculation of the statistics. In a reply to a similar comment by Reviewer #1, we elaborate further about potential influences of directional and velocity wind shear on the correlation.

21. L 435-436: "in order to make measurements better comparable to precipitation data by other instruments". I agree, but then your definition should be comparable with the more common definitions of Dmax used by those instruments. Or at least you should show and describe possible differences (see also major comments).
—> As described under the major comments, we showed the similarity of dmax and dc. Therefore, we believe that the statement is appropriate here.

22. L 461: I can imagine that wind deteriorates the 2DVD calibration. But why is no wind-shield used during calibration? The calibration "table" has no side walls that could hold off wind coming from the side
—> As mentioned under the major comments, another calibration was performed under wind-free conditions, which improved the results a lot.